

# Numerical dispersed flow simulation of fire-flake particle dynamics and its learning representation

Jong-Hyun Kim[1] and Jung Lee[2]

[1] College of Software and Convergence (Department of Design Technology), Inha University, Incheon, Michuhol-gu, Republic of South Korea
[2] Department of Computer Engineering, Hanbat National University, Yuseong-gu, Daejeon, Republic of South Korea

## ABSTRACT

In this article, we propose methods for simulating the detailed flow of dispersed fire-flake particles in response to the movement of a flame, using chaotic advection and various buoyant flow techniques. Furthermore, we utilize these techniques to gather a synthetic dataset of detailed fire-flake particles and extend the solver to represent the movement of fire-flake particles based on learning-based approaches. Fire-flake particles not only exhibit unique and complex movements on their own, but they are also significantly influenced by the movement of the flame and the surrounding airflow. Modeling the flow of fire-flake particles realistically is challenging due to their chaotic and constantly changing nature. Instead of explicitly modeling the complex fire-flake particles in the flame based on fluid mechanics, this article efficiently approximates the chaotic motion of fire-flake particles using two approaches: 1) chaotic advection to simulate the flow and 2) controlled buoyant flow, which varies based on the temperature and lifespan of the fire-flake particles. Additionally, we collect a fire-flake dataset through this simulation and extends the solver to learn the representation of fire-flake motion using neural networks. During the advection process of fire-flake particles, a new stochastic solver is used to calculate the subgrid interactions between them. In this article, not only we propose algorithms that can express these techniques through numerical simulation, but we also extend this solver using artificial intelligence techniques to enable learning representation. By using the proposed technique, it is possible to efficiently simulate fire-flake particles with various movements in chaotic regions, and it allows for more detailed representation of fire-flake particles compared to existing methods. Unlike the typical random walk approach that adds noise randomly to the movement, our method considers the size and direction of the flame. This allows us to express fire-flake particles stably in most scenes without the need for parameter adjustments.

## INTRODUCTION

In various fire-related scenes, such as objects of different materials burning, engulfed by flames, or leading to a bonfire, fire-flake particles appear. Particularly in movies,

Corresponding author
Jung Lee, airjung@gmail.com

commercials, animations, and similar media, when flames rise upwards, numerous fire-flake particles move in a complex and chaotic manner. Both flames and fire-flake particles are the result of complex chemical reactions generated by combustion. Macroscopically, the movement of fire-flake particles can be expressed based on fluid mechanics, including buoyancy, air resistance, turbulence, and diffusion (see Fig. 1).

Recently, there has been a consistent emergence of techniques aiming to learn and represent computationally expensive fluid simulations using neural networks (*Li & Farimani, 2022*; *Tumanov, Korobchenko & Chentanez, 2021*; *Ma et al., 2018*). However, not only fluid simulations, but also secondary effects such as air bubbles, splashes, foam, and fire-flakes, which are represented by the movement of underlying fluids, become even more computationally demanding and challenging to learn. The movement of underlying fluids such as water and fire is typically represented using a grid-based Eulerian approach as an approximation. However, training neural networks on a 3D grid requires a significant amount of memory, making it challenging to generate high-resolution test results.

Fire-flake particles, in general, exhibit complex and unpredictable dispersion or movement at a microscopic level. This characteristic is also observed in air bubbles represented within water. For air bubbles, they are represented by the flow within the water and often focus on capturing the swirling motion that arises from the interaction between the bubbles. On the other hand, fire-flake particles, being lighter than air and significantly influenced by air resistance, exhibit relatively more chaotic and disorderly movement. The complexity of fire-flake particle movement makes the simulation of dispersed fire-flake flow even more challenging.

Methods combining the Eulerian surface tracking framework with multiphase fluid solvers have been proposed to simulate the movement of particles. These techniques have been utilized in the field of computer graphics to represent bubbles and fire-flake particles (*Hong & Kim, 2003*, *2005*; *Song, Shin & Ko, 2005*; *Zheng, Yong & Paul, 2009*). However, the methods developed so far have represented the movement of fire-flake particles at a relatively macroscopic level, mainly due to the numerical limitations of grid-based solvers. To address this issue, a particle-particle interaction model based on smoothed particle hydrodynamics (SPH) has been proposed (*Müller et al., 2005*; *Cleary et al., 2007*; *Hong et al., 2008*; *Mihalef, Metaxas & Sussman, 2009*). Indeed, SPH involves direct computation of particle-particle interactions, making it inefficient for simulating fire-flake particles that are influenced by air resistance in chaotic regions. This problem not only affects computational efficiency but also impacts the stability of the system, making it challenging to utilize in high-quality real-time applications.

To reduce the computational burden, a multi-layer framework is sometimes employed (*Geiger et al., 2006*). In this approach, multiple layers are added to represent effects such as spray, mist, and foam. This technique is commonly utilized in real-time applications like games and virtual reality (*Qiu et al., 2017*). However, since most of the approaches utilize textures for each layer, the movement and rendering quality are limited, and it becomes challenging to control them in a physically accurate manner. Moreover, the movement of dispersed fire-flake particles is influenced by various interactions, such as particle-particle

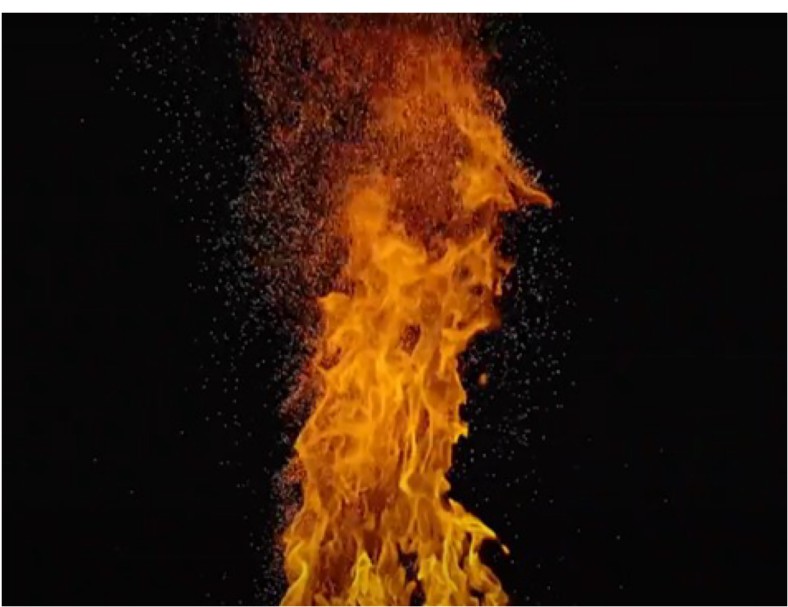

**Figure 1 An example of actual flames and fire-flakes.**

interactions, particle-flame interactions, air resistance, and movement within chaotic regions. Thus, manually expressing these complexities is challenging for users.

In this article, we propose a novel framework to simulate the flow of dispersed fire-flake particles in chaotic regions realistically and efficiently. Furthermore, we introduce a learning component to the framework, enabling the system to learn the representation of fire-flake particle movement. As mentioned earlier, the challenge in simulating such flows lies in accurately representing the movement influenced by air resistance and the disorderly motion observed in chaotic phenomena such as turbulence in multiphase fluids. Additionally, calculating the interactions between fire-flake particles and integrating the physical momentum from subgrid particles into the fluid solver is also a challenging problem. There are approaches that model turbulence represented by the free surface (*Kim, Tessendorf & Thuerey, 2013*; *Mercier et al., 2015*) or model the Rayleigh-Taylor instability occurring when denser material penetrates into less dense material (*Braileanu et al., 2021*). However, relying solely on these methods may not be sufficient to accurately compute the detailed movement of fire-flake particles as observed in flames. Due to the inherent difficulty in directly simulating and representing the movement of fire-flake particles through governing equations, this article proposes a chaotic advection approach utilizing various buoyant flows based on the temperature and lifespan of fire-flake particles. In this article, instead of explicitly computing the boundary conditions between different substances like flame and fire-flake particles, the average movement of dispersed fire-flake flow is modeled using buoyancy based on a continuum approach. Our method utilizes a grid-based solver that allows us to capture the interactions between fire-flakes as well as the interactions between fire-flakes and the flame. Furthermore, to simulate the complex interactions between fire-flake particles, we simulate the subgrid dynamics of the

flame. We introduce a novel probabilistic approach specifically designed for this purpose. In addition to the simulation approach, we extend the solver through the use of artificial intelligence to learn and utilize the movement of fire-flake particles. By employing machine learning techniques, the solver becomes more accessible, allowing users to easily utilize it without needing in-depth knowledge of complex numerical analysis techniques.

The proposed method can be applied in real-time to detailed fire simulations for visual effects (VFX) used in films, games, and metaverse content. Additionally, similar to how previous methods applied flame simulation to wildfires, our approach can more accurately model the propagation of wildfires due to fire-flakes (*Hädrich et al., 2021*; *Kokosza et al., 2024*). Furthermore, as it can track the movement of flames and fire-flakes when human intervention occurs during wildfire suppression, this method could potentially facilitate analysis for predicting and combating wildfires more effectively.

In this article, the calculation of fire-flake particle movement is conducted through four main processes: 1) To efficiently represent the global movement of fire-flake particles, we utilize a continuum solver for the flame. 2) We employ a probabilistic chaotic advection technique to represent the disorderly motion characterized by complex movements. 3) Based on the temperature and lifespan of fire-flake particles, we control the buoyant flow to accurately capture the detailed scattering of fire-flake particles in the air. 4) To facilitate learning, we construct a synthetic dataset using the new simulation approach and extend the technique to encompass the generation, advection, removal of fire-flake particles, allowing for a learning representation of their movement.

In previous approaches, fire-flakes were advected solely by the underlying flow of flame motion. However, this approach alone is insufficient to reproduce the chaotic motion of fire-flakes accurately. To address this limitation, external forces such as vorticity confinement can be applied to particle interactions to artificially create vortex-like behavior. However, this method merely simulates simple particle interactions and fails to fully capture the irregular turbulence effects observed in real fire-flake motion. Additionally, the movement of fire-flakes is influenced not only by flame motion but also by the surrounding airflow, exhibiting a combination of buoyant rising motion and dispersion in the air. This fundamental difference distinguishes fire-flake particles from air bubbles commonly seen in water simulations. To effectively represent these characteristics, this study introduces varying buoyant flows and chaotic advection techniques, proposing a modeling approach that integrates numerical simulation with learning representation.

## RELATED WORK

In this section, we explore the representation of various materials and fire simulations using multiphase fluid simulations.

### Multiphase fluid simulations

*Hong & Kim (2003)* introduced the Volume-of-Fluid (VOF) method to simulate bubble dynamics. Following that, they utilized the ghost fluid method to capture the discontinuity represented by physical properties at the bubble-liquid interface (*Hong & Kim, 2005*). *Song, Shin & Ko (2005)* proposed a novel multiphase fluid solver to alleviate numerical

dissipation issues that arise in gas-liquid simulations. *Mihalef et al. (2006)* applied the Coupled Level-Set and Volume-of-Fluid (CLSVOF) method, initially proposed by *Sussman (2003)* to simulate boiling water. *Kim & Carlson (2007)* also proposed a framework for simulating boiling water in a simplified manner. *Zheng, Yong & Paul (2009)* developed a regional level-set method to capture extremely thin bubble interfaces. *Kim et al. (2007)* expanded on this research and proposed a volume control method for bubbles and foam. *Müller et al. (2005)* developed a multiphase fluid solver based on the SPH method. *Cleary et al. (2007)* proposed a technique for simulating bubbles and the liquid. The aforementioned methods primarily focused on larger bubbles or dynamics related to the bubble-liquid interface.

To capture the movement of smaller bubbles or fire-flake particles that are smaller than the grid resolution, a hybrid approach can be employed, combining grid-based solvers with the use of particles. *Greenwood & House (2004)* utilized the particle level-set method (*Enright, Marschner & Fedkiw, 2002*) to handle escaped particles, which are particles dispersed in the air. They converted these escaped particles into smaller bubbles. *Hong et al. (2008)* extended the representation of escaped particles in a hybrid framework by utilizing SPH method instead of discarding them. Similarly, *Thürey et al. (2007)* combined SPH bubbles with a shallow water framework to simulate fluid behavior. *Mihalef, Metaxas & Sussman (2009)* introduced a method for integrating a particle model into a grid-based simulation using a marker level-set approach. Most existing methods have focused on representing bubbles in water, and there has been relatively less research specifically dedicated to modeling fire-flake particles represented by flames.

## Fire simulations

In physics-based simulations, various methods have been proposed to represent realistic flames. After the introduction of grid-based fluid simulations, similar approaches have been proposed to compute the temperature field and represent flames (*Melek & Keyser, 2002*). *Nguyen, Fedkiw & Jensen (2002)* developed a novel modeling technique to represent the velocity and pressure of flamesat the interface between combusted material and fuel. *Hong, Shinar & Fedkiw (2007)* proposed a method to represent realistic flame patterns by solving detonation shock dynamics based on curvature and coupling it with fluid simulation.

*Horvath & Geiger (2009)* proposed a framework for fast representation of flame simulations in screen space, leveraging the power of GPUs. *Kim, Lee & Kim (2016)* proposed a method for controlling the movement of flames to achieve a target shape by manipulating the temperature. There are studies that focus on representing visual secondary effects resulting from explosions (*Kawada & Kanai, 2011*; *Kwatra, Gretarsson & Fedkiw, 2010*). The methods mentioned earlier for simulating flames can generate velocity and temperature fields that can be used for rendering, resulting in realistic visual effects.

Representing subgrid details in grid-based simulations is a critical research topic, and the utilization of particles is an important approach to achieve this. *Feldman, O'brien & Arikan (2003)* utilized particle-based methods to represent suspended particle explosions resulting from detonations. *Hong et al. (2008)* proposed a particle-grid hybrid technique

that can realistically represent the movement of air bubbles in water. In particular, they introduced an SPH-based vorticity confinement technique, which enhances the realism of air bubble movement (*Hong et al., 2008*). *Zhao et al. (2009)* implemented a fireworks animation that moves according to a specified target shape using GPU-based techniques. *Kim et al. (2012)* proposed a technique for controlling the path of moving air bubbles using a sketch-based approach. *Son et al. (2013)* proposed a still-frame simulation technique for representing fire effects as depicted in images. They focused on creating realistic fire effects based on a given image but did not specifically address fire-flake effects. *Kim et al. (2017)* aimed to model fire-flake particles that are influenced by the movement of the flame in a manner similar to air bubbles. They simulated the movement of fire-flake particles using a method similar to *Hong et al.*'s *(2008)* bubble dynamic. Despite the distinct differences in the movement of fire-flake particles and bubbles, the results showed little disparity. *Kim & Lee (2019)* proposed a framework that can analyze the flow of flame in a 2D image and generate real-time fire-flake effects, which can be rendered in an interactive manner. *Choi et al. (2021)* applied *Kim et al.*'s *(2017)* method to train an artificial intelligence model, but the results showed fire-flake particles that were similar to a basic particle system rather than capturing the complexity of actual fire-flake particles. These simulated fire-flake particles were not sufficient for accurately depicting the intricate motion of fire-flake particles dispersed in the air. Recently, *Nielsen et al. (2019)* proposed a physics-based combustion simulation technique. While this method is comprehensive in modeling various aspects of combustion, such as fuel combustion, chemical kinetics, radioactive heating, flame propagation, soot formation, and oxidation, it does not explicitly address the representation of fire-flake effects.

Recent studies have proposed several methods related to fire and flame simulation. *Nielsen et al. (2019*, *2022)* introduced a physics-based combustion simulation technique. This method represents flames, temperature, and soot distribution more realistically than previous approaches. It proposes a mathematical model for the thermodynamic properties of real-world fuels, enabling predictions of adiabatic flame temperatures. Additionally, it introduces a new heat transfer model that incorporates convection, conduction, as well as radiative cooling and heating.

*Hädrich et al. (2021)* proposed a novel wildfire simulation approach aimed at realistically representing the combustion process of individual trees and the resulting fire propagation on a forest scale using flame simulation. They modeled each plant as a 3D geometric model and suggested a combustion process for the plant considering effects like heat transfer, char insulation, and mass loss. *Kanyuk et al. (2023)* presented a method to control flame silhouettes using volumetric neural style transfer. Unlike traditional style transfer, this method enhances the expressiveness of fire simulation for character animation by exaggerating the visual appearance of flames. *Kokosza et al. (2024)* introduced a novel approach to simulate the dynamic interactions among various components of wildfires, including convection, combustion, and heat transfer processes between vegetation, soil, and atmosphere. They also modeled fire ignition, generation, and transport caused by embers, allowing for the simulation and rendering of ember transport by wind, the impact of ground fires, and the effect of fire barriers.

The methods mentioned above aim to represent natural flames or use style transfer to exaggerate flame shapes for animation. While there have been studies utilizing flame simulations to analyze and predict natural disasters caused by wildfires, there has been little research focused specifically on the representation of fire-flakes. Since fire-flakes are secondary effects that accompany fire simulations, they involve significant computational demands. This article presents a framework that enhances visual detail and improves efficiency through learning representation, focusing on the fire-flakes themselves.

## PROPOSED FRAMEWORK

### Underlying fluid solver

In this article, the flame is assumed to exhibit the characteristics of an incompressible multiphase fluids flow, and the modeling is based on this assumption. The Incompressible Navier-Stokes equations are expressed as equations of momentum conservation and mass conservation (see Eqs. (1) and (2)).

$$\mathbf{u}_t + (\mathbf{u} \cdot \nabla p/\rho) = \nabla \cdot (\mu \nabla \mathbf{u})/\rho + \mathbf{f}/\rho \tag{1}$$

$$\nabla \cdot \mathbf{u} = 0, \tag{2}$$

where $\mathbf{u}, p, \rho, \mu$, and $\mathbf{f}$ represent velocity, pressure, density, viscosity, and external force, respectively.

Furthermore, the radiational cooling rate, denoted as $C_T T^4$, is modeled as *Nguyen, Fedkiw & Jensen*'s *(2002)* method, where $C_T$ is a user-selectable constant. This model rapidly cools high-temperature regions but cools lower temperature regions relatively slowly. Low-temperature regions should be cooled primarily through the temperature diffusion term, represented as $\tilde{k} \nabla^2 T$, where $k$ is the thermal conductivity coefficient. However, diffusion tends to smooth out temperature details, which may not be suitable for applications aiming to preserve rich details in fire simulations. Therefore, for low-temperature cooling, the Newton's Law of cooling model is used, which also takes into account the exponential decrease of the surrounding temperature in the vicinity ($T = 0$ in our experiments). This is equivalent to replacing the temperature diffusion term $\tilde{k} \nabla^2 T$ with an exponential decay term, denoted as $-d_T T$, where $d_T$ is a user-controllable decay coefficient. Taking all of these factors into account, the modified heat equation, considering both heating and cooling effects, can be expressed as Eq. (3).

$$\frac{\partial T}{\partial t} = -\mathbf{u} \cdot \nabla T + \frac{H}{\rho C_\mu} F_b - C_T T^4 - d_T T, \tag{3}$$

where $H$ represents the heating value of the fuel, indicating the amount of energy released when a unit mass of the fuel is combusted. Gas fuel moves forward along the velocity field and undergoes diffusion as it burns. The fuel behavior is modeled as Eq. (4).

$$\frac{\partial F}{\partial t} = -F_b - \mathbf{u} \cdot \nabla F + \mu_F \nabla^2 F, \tag{4}$$

where $\mu_F$ represents the fuel diffusion coefficient. If the temperature $T$ is lower than $T_{ignition}$, the burn rate $F_b$ becomes 0. In general, $F_b$ is a function of stochastic air/fuel

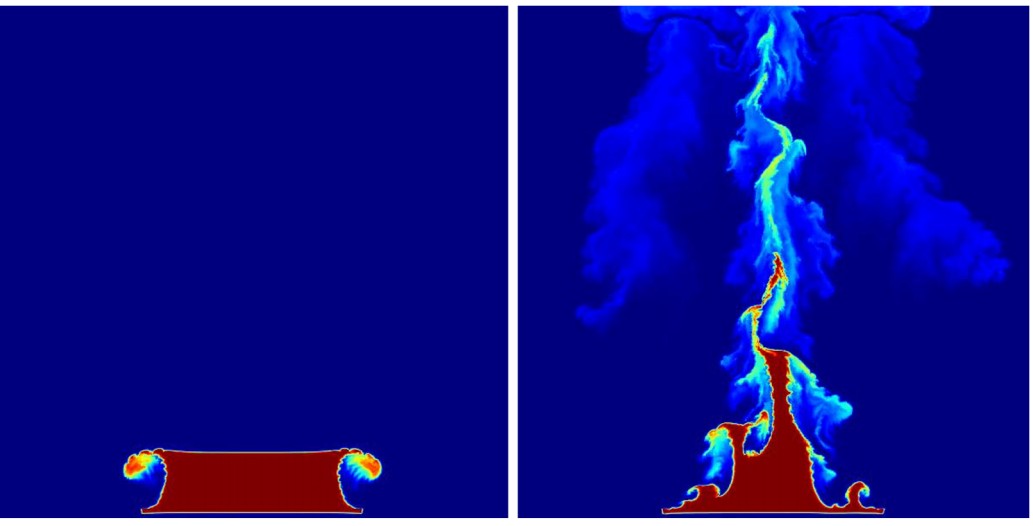

**Figure 2  Underlying fluid simulations.**

mixture, oxygen density, and burn rate of the fuel. In addition, $F_b$ is assumed to be constant, and this value is a user-defined constant used to control the combustion rate of the fuel. In this article, the stable fluids method is employed to compute the above equations (*Stam, 1999*) (see Fig. 2).

### Fundamental force of fire-flake particles

In this article, the forces acting on fire-flake particles are modeled using four different types (*Kim et al., 2017*): airflow force $\mathbf{f}^{air}$, drag force $\mathbf{f}^{drag}$, lift force $\mathbf{f}^{lift}$, and buoyancy force $\mathbf{f}^{buoy}$. Each force is applied to each fire-flake particle to update its velocity (see Eq. (5)).

$$\mathbf{f} = \mathbf{f}^{air} + \mathbf{f}^{drag} + \mathbf{f}^{lift} + \mathbf{f}^{buoy}. \tag{5}$$

The most significant factor influencing the movement of fire-flake particles is airflow. As defined in Eq. (5), the velocity field generated in the simulation space can realistically represent the movement of fire-flake particles through vorticity control. Firstly, we compute the airflow $\mathbf{f}^{air}$ as follows, according to Eq. (6):

$$\mathbf{f}_i^{air} = k_{air}\mathbf{v}_i, \tag{6}$$

where $k_{air}$ is the coefficient related to the airflow velocity, and $\mathbf{v}_i$ represents the interpolated velocity obtained from the grid velocity, $\mathbf{u}$. The interpolated position becomes the position of a fire-flake particle.

Due to the very small mass of fire-flake particles, they exhibit chaotic movement in the air. This kind of motion plays a crucial role in realistically representing fire-flakes and their behavior. *Hong et al. (2008)* have utilized drag and lift forces to model the movement of water droplets, and in this article, we adapt their method to suit the movement of fire-flake particles (see Eq. (7)). In the air, the drag force acts as a resistance opposing the movement of fire-flake particles, while the lift force is generated perpendicular to the surrounding

airflow. This process is calculated using the velocity field and the temperature of the fire-flake particles.

$$\mathbf{f}_i^{drag} = -k_{drag} r^2 |\mathbf{v}_i - \mathbf{u}_i| (\mathbf{v}_i - \mathbf{u}_i), \tag{7}$$

where $k_{drag}$ represents the drag coefficient, $\mathbf{v}_i$ is the velocity of the fire-flake particle, $\mathbf{u}_i$ is the grid velocity, and $r$ is the radius of the fire-flake particle defined by the user.

The lift force is defined as Eq. (8).

$$\mathbf{f}_i^{lift} = -k_{lift} V_i T_i (\mathbf{v}_i - \mathbf{u}_i) \times \omega_i, \tag{8}$$

where $k_{lift}$ is the lift coefficient, $V_i$ is the volume of the fire-flake particle, $T_i$ is the temperature of the fire-flake particle, and $\omega_i$ is the vorticity of the airflow around the fire-flake particle: $\omega_i = \nabla \times \mathbf{u}_i$. The temperature of the fire-flake particle is obtained by interpolating the grid temperature values.

The buoyancy force on the fire-flake particle is calculated considering the temperature, as shown in Eq. (9).

$$\mathbf{f}_i^{buoy} = k_{buoy} T_i \mathbf{f}_{buoy}, \tag{9}$$

where $k_{buoy}$ is the buoyancy coefficient, and $\mathbf{f}_{buoy}$ represents the upward normalized vector.

**Limitations of this approach.** The above method is a modified algorithm based on *Hong et al.*'s *(2008)* approach, *Kim & Lee (2019)*, incorporating temperature considerations. This method effectively represents fire-flake particles moving according to the airflow. However, unlike air bubbles in water, near the flame, there are turbulent interactions between fuel and air that give rise to chaotic movements. The above method may not be sufficient to accurately represent these intricate details. The complex airflow caused by the flame sometimes exhibits movements that are not fully captured by buoyancy alone, as it also involves viscous behavior. Fire-flake particles, in addition to being influenced by the air, exhibit various velocities and momenta, which can sometimes result in a velocity field that appears to include viscosity-like effects.

## Generating fire-flake particles

Generally, fire-flake particles are generated by external forces applied to carbon-based fuels or through the injection of air. Small pieces that detach from carbon-based fuels are hot enough to self-combust and light enough to float in the air, appearing as small red spots known as fire-flake particles. In the boundary region where fire-flake particles are generated, there is relatively high volatility due to changes in temperature and velocity. In this article, an energy function is used to predict locations where the temperature is sufficiently high and where there is a possibility of abrupt changes in velocity and temperature (see Eq. (10a)).

$$E_i = E_i^t + E_i^k \tag{10a}$$
$$E_i^t = k_t T_i \tag{10b}$$
$$E_i^k = \frac{1}{2} \rho \mathbf{v}_i^2, \tag{10c}$$

where $E^t$ represents thermal energy, $k_t$ is the heat capacity coefficient, $E^k$ denotes kinetic energy, and $\rho$ stands for density. As a result, in regions where the temperature exceeds kappat and the temporal energy change surpasses kappae, there is a higher likelihood of fire-flake particle generation. To identify such regions, we utilize Eq. (11).

$$\frac{\partial E}{\partial t} > \kappa_e \wedge E^t > \kappa_t, \tag{11}$$

where $\kappa_t$ and $\kappa_e$ are user-specified threshold values, and in this article, we set them to 2.0 and 0.6, respectively, based on multiple experiments. All the regions identified through Eq. (11) will generate fire-flake particles, but the ones that fall outside the specified generation range need to be handled separately. By modifying $\kappa_{t,e}$, this problem can be alleviated, and this approach uses a randomization method to avoid a monotonous generation pattern. This approach is similar to the method proposed by *Kim et al. (2017)* (see Eq. (12)).

$$\mathbf{C} = \left\{ (i,j,k) | C_{cons} > \delta, (i,j,k) \in \mathbb{R}^{N \times N} \right\} \tag{12a}$$

$$C_{cons} = R\left( \frac{E^k E^t}{\max(E_0^k, \dots E_{N-1}^k) \max(E_0^t \dots E_{N-1}^t)} \right), \tag{12b}$$

where $R$ is a uniformly distributed random number in the range [0,1], and $\delta$ is a user-specified threshold. In this article, delta is set to 0.15.

## Varying buoyant flows to preserve detail

In this section, we aim to represent the influence of turbulence caused by the airflow, such as multiphase fluids, by modeling a new form of buoyancy (see Eqs. (13) and (14)).

$$\mathbf{f}_i^{buoy'} = k_{buoy} T_i \mathbf{f}_{buoy} \eta \tag{13}$$

$$\eta_{t+\Delta t} = \eta_t + \frac{1}{2} m ||\mathbf{v}||^2 - \varepsilon, \tag{14}$$

where $\eta$ represents the lifespan of fire-flake particles, and $\varepsilon$ is the damping coefficient. $\eta$ is integrated with the buoyancy to model various forms of buoyancy fields. In this article, $\eta$ is set to 0.04. Generally, the lifespan of fire-flake particles is constant. However, in this article, a different method for its calculation is employed: The design involves accumulating the kinetic energy of fire-flake particles while subtracting by a threshold value $\varepsilon$ (more detailed explanation is mentioned in "Proposed Framework"). In this article, buoyancy has been adjusted to be proportional to the temperature and lifespan of fire-flake particles. By doing so, the buoyancy was realistically modeled, resulting in more accurate movement compared to the previous approach that treated buoyancy as a constant. In chaotic regions, particles exhibit a wide range of velocities and momenta. Therefore, in this article, the buoyancy magnitude is controlled through the temperature and lifespan to represent these characteristics. As a result, fire-flake particles with low temperature and short lifespan have minimal momentum and a high likelihood of disappearing. Therefore, their movement is characterized by a minimal application of buoyancy.

In addition, this article introduces the following forces to capture the motion of fire-flake particles scattering in the air due to turbulence (see Eq. (15)).

$$\mathbf{f}_{i,j}^{vorticity} = \varepsilon \left( \mathbf{N} \times \frac{\omega}{|\omega|} \right) \rho_i \eta_i T_i. \tag{15}$$

The vorticity is calculated at the midpoint between two adjacent fire-flake particles: $\omega = \nabla \times \mathbf{v}$. The position vector of the vorticity is calculated as follows: $\eta = \frac{m_i \mathbf{p}_i + m_j \mathbf{p}_j}{m_i + m_j} - \mathbf{p}_i$, this value is normalized before being used: $\mathbf{N} = \frac{\eta}{|\eta|}$. As a result, the vorticity is expressed more strongly when the temperature of fire-flake particle is high and its lifespan value is large.

## Chaotic advection for subgrid fire-flake dynamics

In this section, we propose a probabilistic solver to implement the chaotic advection of fire-flake particles. Many natural phenomena, such as cloud motion, wave turbulence, ink droplets diffusing in water, and the movement of fire-flake particles, exhibit chaotic behavior. In this article, Gaussian random numbers are used to model the advection of fire-flake particles, aiming to represent these chaotic movements and behaviors. Rising fire-flake particles create waves and interact with other particles in the air, generating chaotic fluid dynamics. In theory, accurately capturing the vortices and flow generated by each fire-flake particle would be ideal. However, this approach can be computationally inefficient due to the consideration of the diverse environmental factors and high-resolution micro fire-flakes that occur in chaotic regions. Instead, we model this phenomenon using a discrete random walk. As shown in Fig. 1, fire-flake particles are scattered due to turbulence generated between the fuel and air. To model this process, a probability function $s$ is used to represent the likelihood of dispersed movement. This function is defined as Eq. (16).

$$s_i = \varphi(1 - T_i)\mathbf{v}_i + \delta \mathbf{g}_i, \tag{16}$$

where $\varphi$ is the user-specified dispersion coefficient which is set to 0.25, $\delta$ is the weighting coefficient for the random walk, and $1 - T_i$ represents the gas fraction. After updating the velocity for each fire-flake particle, the value of $s$ is measured. Due to the individual temperature of each fire-flake particle being relatively lower than the grid temperature, $\mathbf{g}_i$ is introduced to track the movement of fire-flake particles dispersed into the air. When fire-flake particles disperse into the air, their velocity and temperature become very weak. In this case, $\mathbf{g}_i$ can be used to compute a new direction for their movement. This can be represented as Eq. (17), which depicts a probabilistic random number based on Gaussian random.

$$\mathbf{g}.\mathrm{x} = \frac{g_{rand}}{w - 1}, \mathbf{g}.\mathrm{y} = \frac{g_{rand}}{w - 1}. \tag{17}$$

In general programming languages, there is a built-in pseudo-random number generator that generates numbers at regular intervals. Most generators use the linear congruential or power residual methods to represent pseudo-random numbers in the

range between 0 and 1, and they are widely used in many researches. In this article, the following steps are taken to calculate the probabilistic solver: 1) Seed initialization, 2) Generation of Gaussian random numbers, and 3) Calculation of probabilistic movements.

In the seed initialization process, the variables required for Gaussian random number generation are initialized using Eq. (18).

$$g_{addition} \leftarrow \sqrt{3\zeta}, g_{fractal} \leftarrow \frac{2g_{addition}}{\zeta v}, \tag{18}$$

where $v$ represents RAND_MAX, and $\zeta$ is the weight for the random number, which is set to 4 in this article.

Gaussian random numbers are generated using Eq. (19).

$$g_{rand} = g_{fractal}\left(\sum_{i=0}^{\zeta} \text{rand}()\right) - g_{addition}, \tag{19}$$

where $\zeta$ is the weight value for the previously set random number.

We utilize Eq. (16) to set different conditions for advecting fire-flake particles, depending on the specific requirements. 1) If the magnitude of the velocity of fire-flake particles is greater than the threshold value of 0.1, we set $\delta$ to 50. 2) If the magnitude of the acceleration of fire-flake particles is greater than the threshold value of 0.2, we set delta to 5. Finally, $s_i$ is added to $\mathbf{v}_i$, and we update the position through Euler integration.

Figure 3 shows the result of applying $s$ when fire-flake particles move in the $X$-axis direction due to external forces. The zigzag patterns of fire-flake particles representing chaotic behavior are well-captured. The results of this method are similar to the air bubble representation proposed by *Kim, Song & Ko (2010)*. In the previous method, a probabilistic approach was used to simulate the movement of micro-scale bubbles, but it resulted in an excessive representation of zigzag motion, leading to noisy results. Our method better represents the characteristics of fire-flake, such as the chaotic movement due to air resistance, compared to the previous technique (see Fig. 3). When attempting to achieve a more accurate numerical approximation of the complex movement of fire-flake particles, not only does the computational workload increase significantly, but numerical instability can also arise during the discretization process, leading to frequent cases of simulation failure. In order to improve computational efficiency and numerical stability, this article utilized a random-based probabilistic model.

## SOLVER EXTENSIONS

In this section, we will describe several extension techniques to represent the simulation algorithm we have discussed so far using artificial intelligence for learning. 1) The generation of fire-flake particles, represented within the grid, is learned through a classification model using artificial neural networks. 2) The movement of fire-flake particles is predicted through learning based on a polynomial regression model. 3) By training the deletion of fire-flake particles using a network-based approach, it becomes easier to control the behavior of fire-flake particles in various scenes. As a result, by using

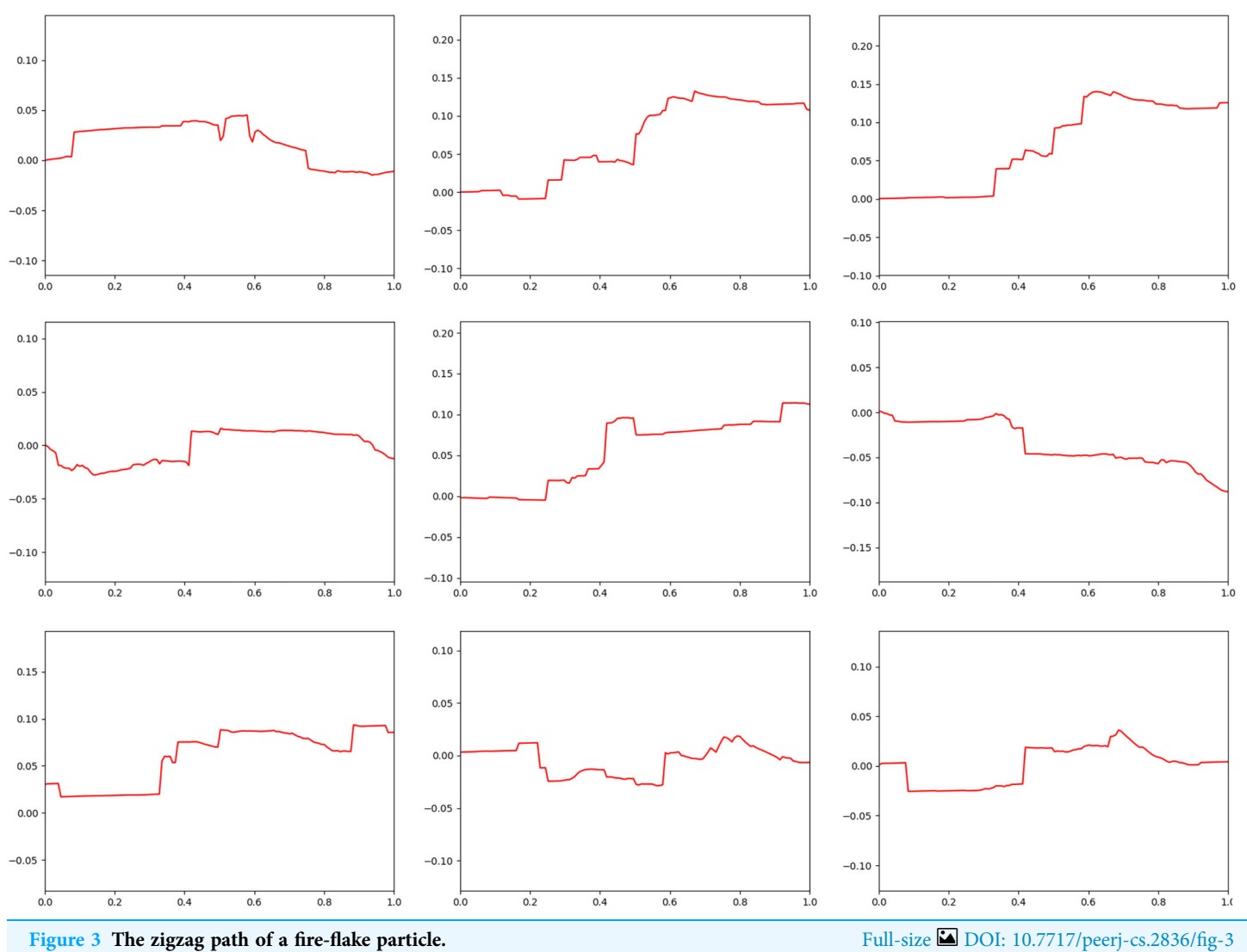

Figure 3 The zigzag path of a fire-flake particle.

the proposed simulation technique, we were able to construct a dataset for fire-flake particles. This approach allowed us to represent fire-flake effects based on the movement of the flame without the need for complex mathematical or physical theories. The experimental results demonstrated the ease and effectiveness of our method in expressing fire-flake effects. In the artificial neural network, the network for fire-flake generation and the network for movement inference are trained independently. The training data undergoes a preprocessing step to be suitable for training the network before being applied to the network. After the training is completed, the test results for each model are visualized.

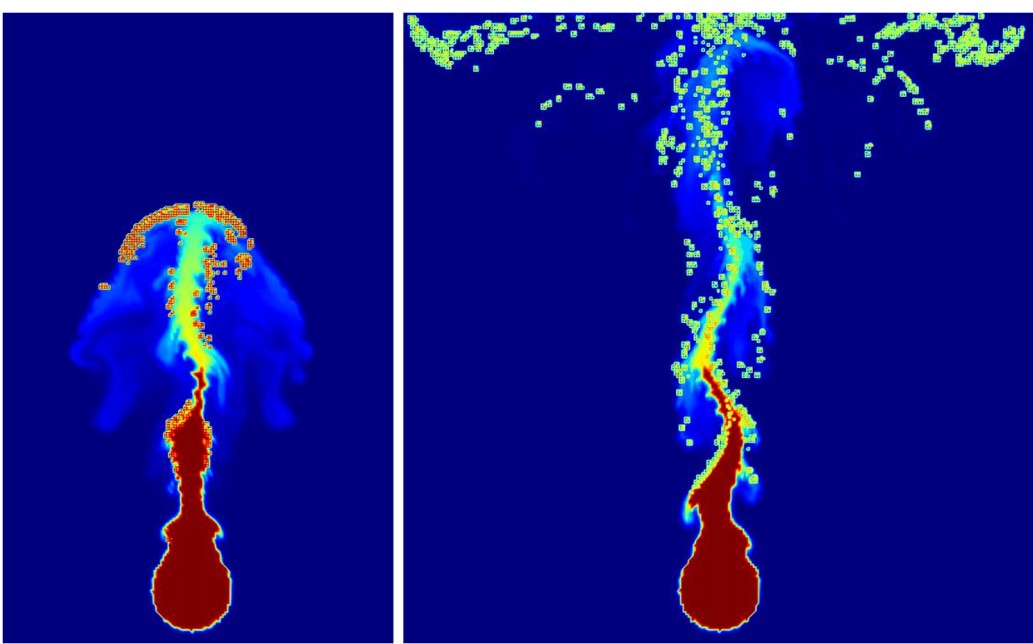

**Figure 4 Regions where fire-flake particles are generated through simulations.**

## Learning for fire-flake generation

In this article, the density, temperature, and velocity obtained from grid-based flame simulation are used for network training to determine whether to generate fire-flake particles. The resolution of the simulation grid for creating the dataset was set to 256 × 256, and for each node, the decision of whether to generate fire-flake particles or not was indicated using a flag. The training is conducted over the entire grid, but in the end, only a very small portion of the nodes actually have fire-flake particles generated compared to the total number of nodes (see Fig. 4). Using such imbalanced data can lead to biased learning and make it challenging to accurately infer the generation of fire-flake particles. To address this issue, in this article, we refine the necessary data for classification training through the ghost cell and 1:1 sampling process.

Ghost cell is a method where padding is added to the grid containing fire-flake particles, and it sets the surrounding grid cells as valid values. However, as mentioned earlier, this can lead to the problem of data asymmetry and ultimately become a hindrance to proper learning. Although using ghost cells allows us to temporarily increase the number of valid cells by acquiring additional surrounding cells, it is still insufficient compared to the total number of nodes. Hence, in this article, we utilize the 1:1 sampling technique to address this issue. By equalizing the number of data samples for each class in the classification model through 1:1 sampling, we prevent biased learning caused by imbalanced data distribution. For example, if there are 350 nodes with fire-flake particles and 900 nodes without fire-flake particles, we use random sampling to adjust the number of nodes without fire-flake particles to 350, achieving a 1:1 ratio between the two classes.

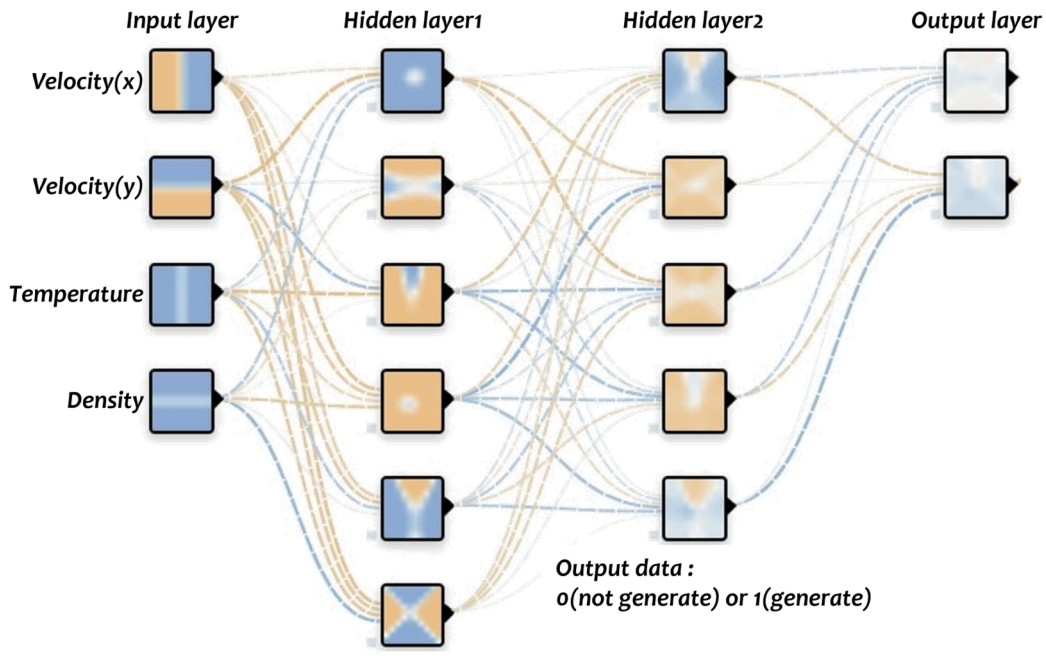

**Figure 5 Generator network of fire-flake particles.**

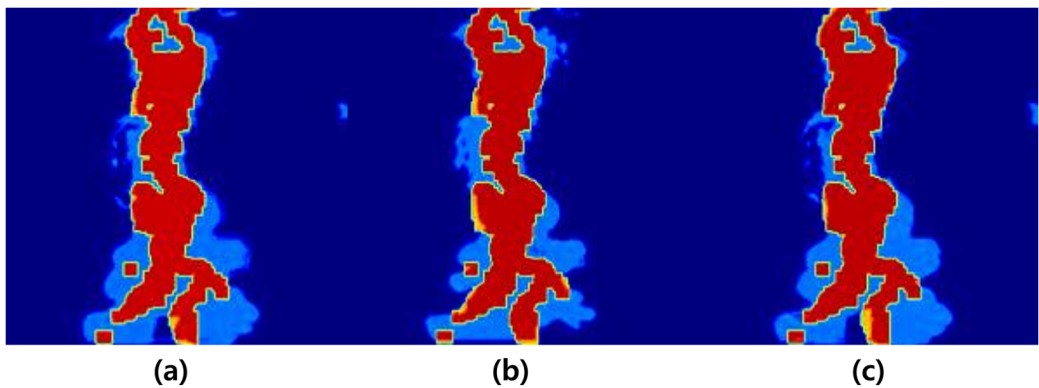

**Figure 6 Generation regions of fire-flake particles predicted using various activation functions (sky blue: flame, red: generation region).** This result is a filtered figure for a clearer visualization: (A) ReLU, (B) Leaky ReLU, (C) Softmax.

 After the preprocessing and refinement process mentioned earlier, the data is used to train the network for fire-flake generation. The fire-flake generation network model consists of four layers, and the activation functions used are Leaky ReLU and Softmax (see Fig. 5). The negative slope used in Leaky ReLU is 0.1, and the results were similar to those obtained using ReLU and Tanh functions. As seen in Fig. 6, the results of the three activation functions are similar. Therefore, in this article, Leaky ReLU and Softmax were used as the activation functions. In Fig. 6, the sky-blue area represents the flame, and the red area represents the inferred positions where fire-flake particles should be generated

using the trained network. Adam was used as the optimizer, and optimization of 9,661 epochs was performed. During this process, the loss was measured to be 0.25.

## Learning for fire-flake advection

In this section, we introduce a network that predicts the movement of fire-flake particles. The proposed model doesn't directly infer the current velocity of fire-flake particles, $\mathbf{v}_i$, for the current frame. Instead, it predicts the change in velocity, $\Delta \mathbf{v}_i^*$, for the fire-flake particles using artificial intelligence. The input feature vector includes the flame velocity of the node where the fire-flake particle exists, as well as the flame velocities of adjacent nodes. Additionally, it incorporates the velocity of the corresponding fire-flake particle. In this case, adjacent nodes refer to the nodes located in the 2D space directly above, below, to the left, and to the right of the current node. The network for fire-flake particle advection has one additional layer compared to the generator network. It employs ReLU as the activation function and uses the least squared error (LSE) as the cost function for updating the weights (see Eq. (20) and Fig. 7).

$$L = \sum_{i=0} \|\Delta \mathbf{v}_i - M_i\|^2, \tag{20}$$

where $\Delta \mathbf{v}_i$ is computed as follows: $\mathbf{v}_{t+\Delta t} - \mathbf{v}_t$. $M_i$ represents the output of the advection network. The difference in velocity changes is applied to the cost function to update the weights.

We determine the generation location of fire-flake particles through the fire-flake generation network. After obtaining the velocity change of the fire-flake particle, $\Delta \mathbf{v}^*$, through the advection network, we then update the position of the fire-flake particle using Euler integration (see Eq. (21)).

$$\begin{aligned} \mathbf{v}_{t+\Delta t} &= \mathbf{v}_t^{\Upsilon} + \Delta \mathbf{v}^* \Delta t \\ \mathbf{p}_{t+\Delta t} &= \mathbf{p}_t + \Delta \mathbf{v}_{t+\Delta t} \Delta t, \end{aligned} \tag{21}$$

where $\mathbf{v}_t^{\Upsilon}$ represents the interpolated velocity from $\mathbf{u}$, and the interpolated position is $\mathbf{p}$. Adam was used as the optimizer, and optimization of 897 epochs was performed. During this process, the loss was measured to be 0.19. Figure 8 represents the fire-flake particles that have been modeled through the training of the two neural networks mentioned earlier. The figure clearly depicts the generated positions and movements of fire-flake particles that have been learned through the training process.

## Learning for fire-flake removal

In the simulations conducted in this study, the lifespan of fire-flake particles was designed by accumulating the particle's kinetic energy while subtracting a threshold $\varepsilon$ (see Eq. (13)). Fire-flake particles with a negative lifespan were removed, resulting in the elimination of particles with low kinetic energy in the process.

The above process can also be controlled through network training. The lifespan of a fire-flake particle is associated with its velocity, so we added a network that takes the velocity and kinetic energy of the fire-flake particle as input vectors. The output of this

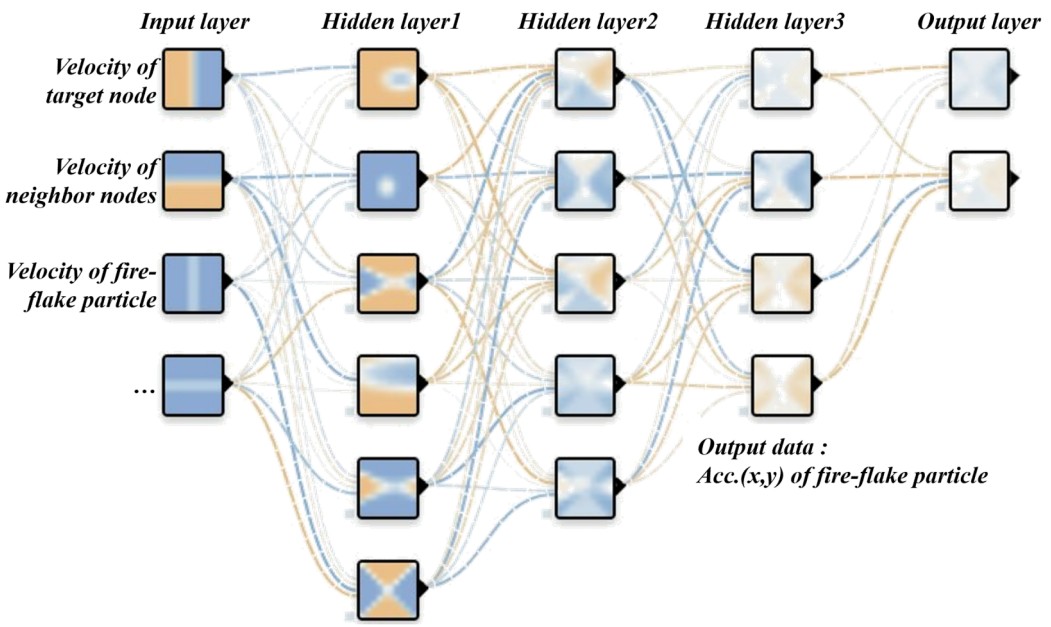

**Figure 7 Advection network of fire-flake particles.**

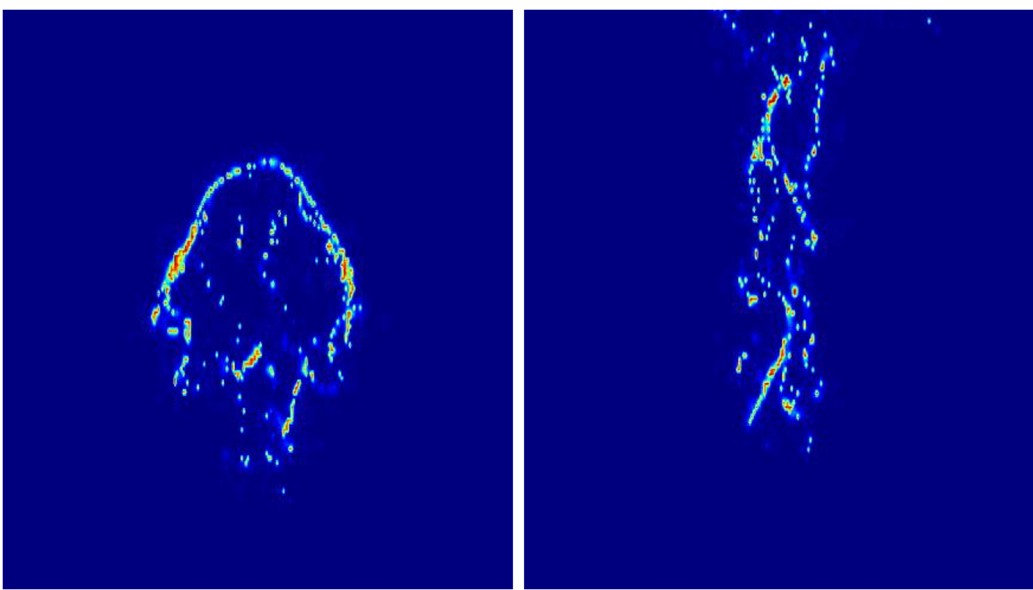

**Figure 8 Generating fire-flake particles in rising smoke simulation with our method.**

network is used to infer the lifespan of the particle. The network architecture is similar to the advection network. In this study, particles with a lifespan below 0 were removed, and by controlling this threshold value, the number of fire-flake particles can be adjusted. In the neural network designed to remove fire-flake particles, the input data includes the velocity of the fire-flake particle ($\mathbf{v}$) and the grid velocity ($\mathbf{u}$), where the grid velocity

represents the velocity of the node to which the fire-flake particle belongs. The target data for the neural network consists of the acceleration and lifespan of the fire-flake particles. In this process, the dataset used to train the lifespan includes the same issue of inhomogeneity that was present in the dataset from the generator network. In this article, similar to the mentioned approach, the active fire-flake particle data and the deleted fire-flake particle data are refined to maintain a 1:1 ratio for network training.

## EXPERIMENT AND RESULTS

To produce the results of this study, we utilized a computer equipped with an Intel Core i7-7700K CPU, 32 GB RAM, and a GeForce GTX 1080Ti GPU. In this article, we proposed a stochastic-based chaotic advection method and various buoyancy flows to efficiently represent the movement of dispersed fire-flake particles by the flame and the detailed fire-flake flow. Furthermore, we extended the solver to enable the representation of these methods through AI-based learning. In this article, we validate the efficiency and the improved visual quality of the proposed method through experimentation and analysis using several scenarios. The experimental results are analyzed from two perspectives: simulation and artificial intelligence.

### Simulation results

In this section, the results generated through the proposed simulation technique are analyzed and compared with previous methods. Figure 9 compares the differences between our method and the previous method in simulating the rising flame driven by buoyancy in the simulation results. A grid resolution of $256 \times 256$ was used to represent the flame, with a time-step of 0.1. Additionally, vorticity confinement (*Fedkiw, Stam & Jensen, 2001*) was applied in the simulation process.

Around the initial frames, approximately at the 37th frame of the simulation, fire-flake particles are generated near the boundaries of the flame. In *Kim et al.*'s *(2017)* method, the temperature of fire-flake particles gradually decreases, leading to the lack of dispersed movement. These particles primarily follow the motion of the flame, resulting in the limited representation of chaotic fire-flake behavior (see Fig. 9B). In our method, even in regions with lower temperatures, the dispersed movement of fire-flake particles is relatively distinct (see Fig. 9A).

Unlike air bubbles, fire-flake particles exhibit highly dynamic movements in chaotic regions. For example, bubbles underwater rise due to buoyancy and can interact with each other, leading to the merging or splitting of bubbles (*Hong et al., 2008*). However, these phenomena often depend heavily on the underlying flow, as they occur underwater. In the previous method as well, the movement of fire-flake particles depended on the flows of underlying fluids, making it challenging to represent the dispersed movement of fire-flakes in the air. Figure 10 illustrates the visualization of fire-flake particles being dispersed by the air in the results generated using our method. The pink particles represent the fire-flake particles dispersed by the air, and in this case, their movement is influenced more by $\mathbf{g}$ than by $\mathbf{v}$.

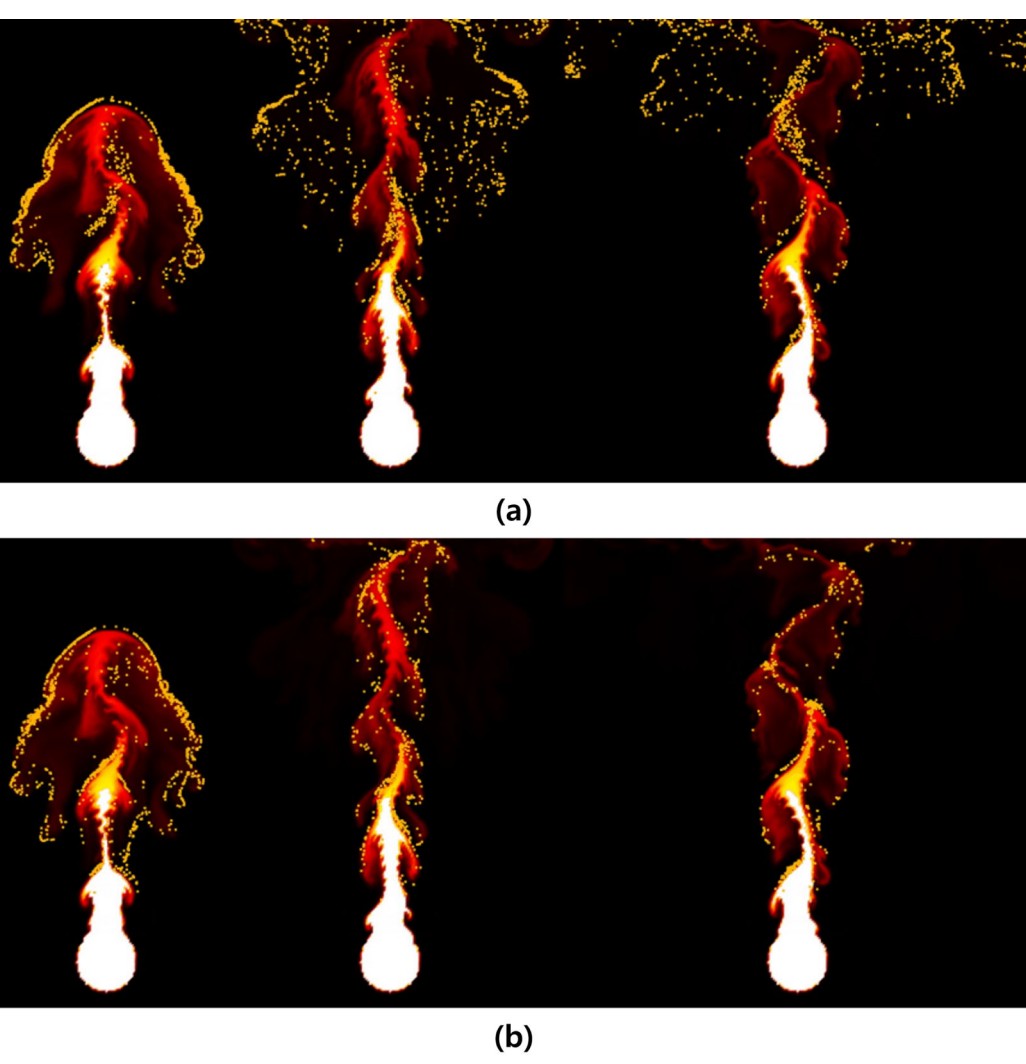

**Figure 9 Comparison of rising-flame results (orange particle: fire-flake, frame number: 37, 65, 92): (A) Our method, (B) Previous method.**

Figure 11 visualizes the velocity of the fire-flake particles. The figure shows a mixture of movements with varying velocities, and this diversity in velocities contributes to capturing the intricate details of the fire-flake particles' behavior. In the previous method, fire-flake particles exhibited similar-sized and directional velocities in adjacent positions. However, in our method, the presence of diverse sizes and directions allows for the representation of chaotic movements, enabling a more realistic results.

Figure 12 illustrates the fire-flake effects generated from the flame's lateral movement when the fuel is not fixed but moves from side to side. The fire-flake particles were generated in accordance with the sharp deformation of the flame's shape due to lateral movement. They effectively captured the upward dispersion as the flame rises. Figure 13 represents the dispersed particles as shown earlier, using colors to visualize their distribution. Our method effectively captured the complexity of motion within the dispersed fire-flake particles even in sharp and thin flame structures.

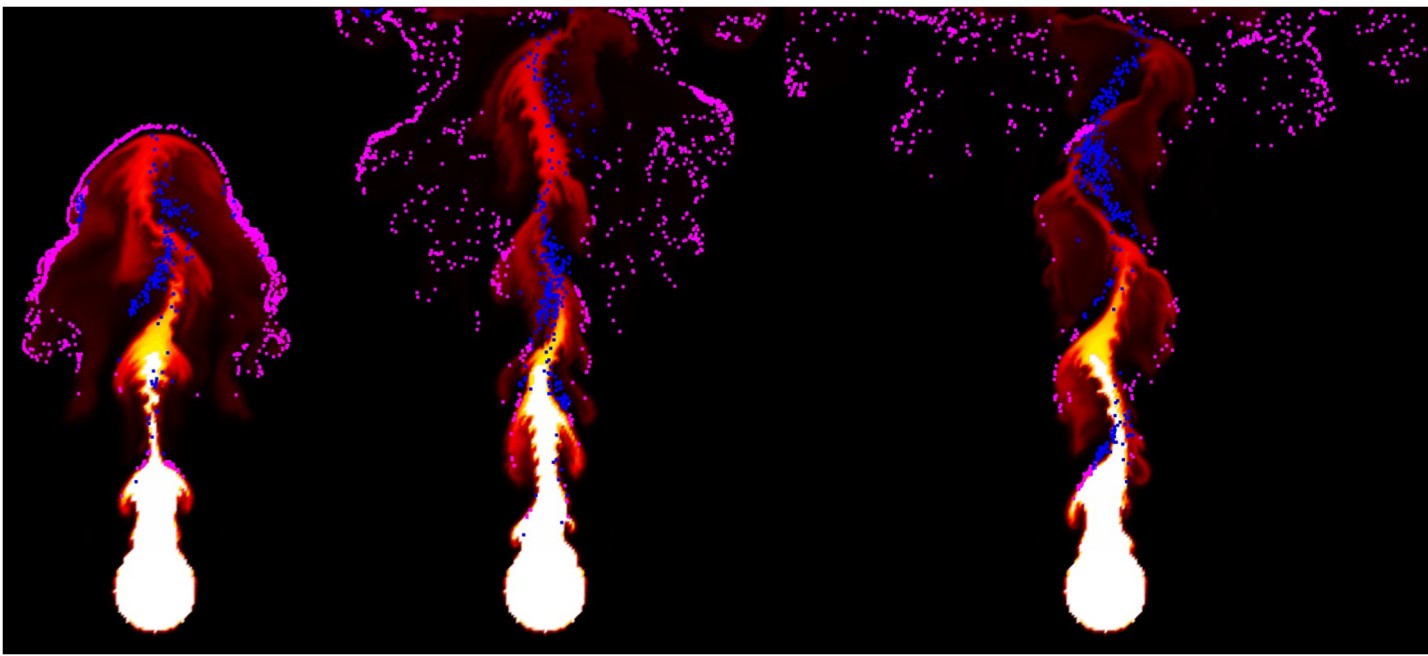

**Figure 10 Fire-flake particles with different movements in chaotic region.**

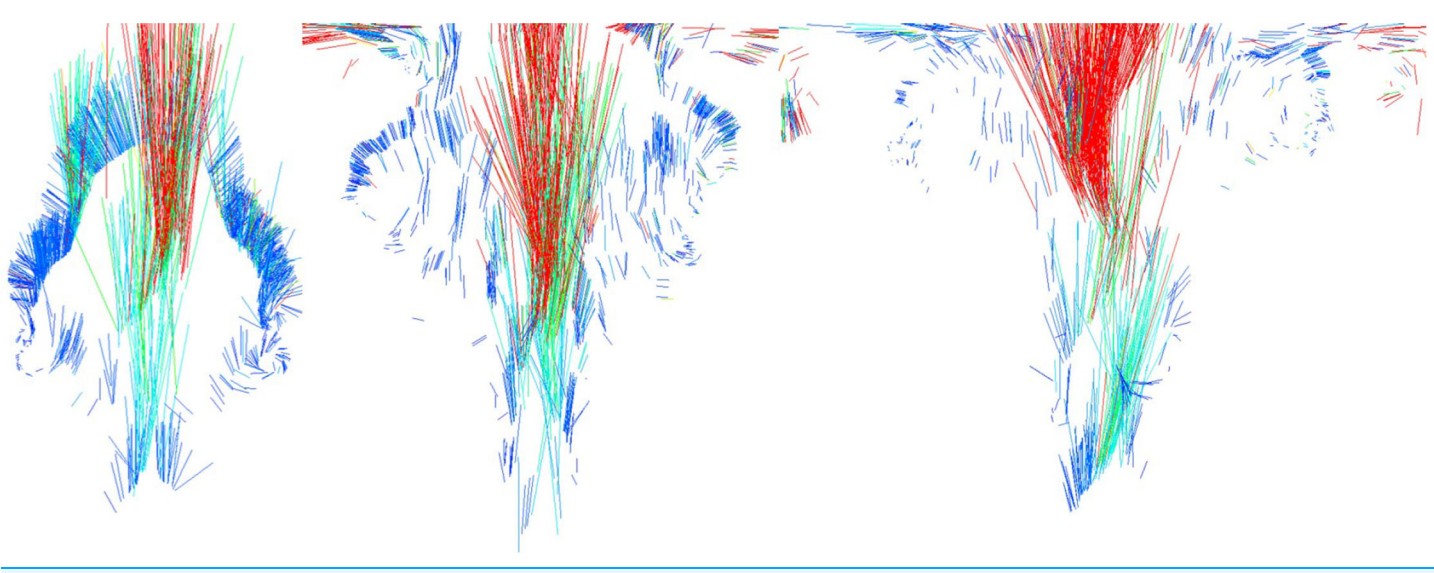

**Figure 11 Visualizing the velocity of fire-flake particles.**

Figure 14 illustrates fire-flake particles generated from a burning flame fueled by a thin layer of fuel, differing from the previous scenarios with thicker flames. In scenes like Figs. 9 and 12, the injected fuel is relatively thick, resulting in a prominent flame. Conversely, Fig. 14 shows a flame that converges towards the center and rises weakly, leading to less

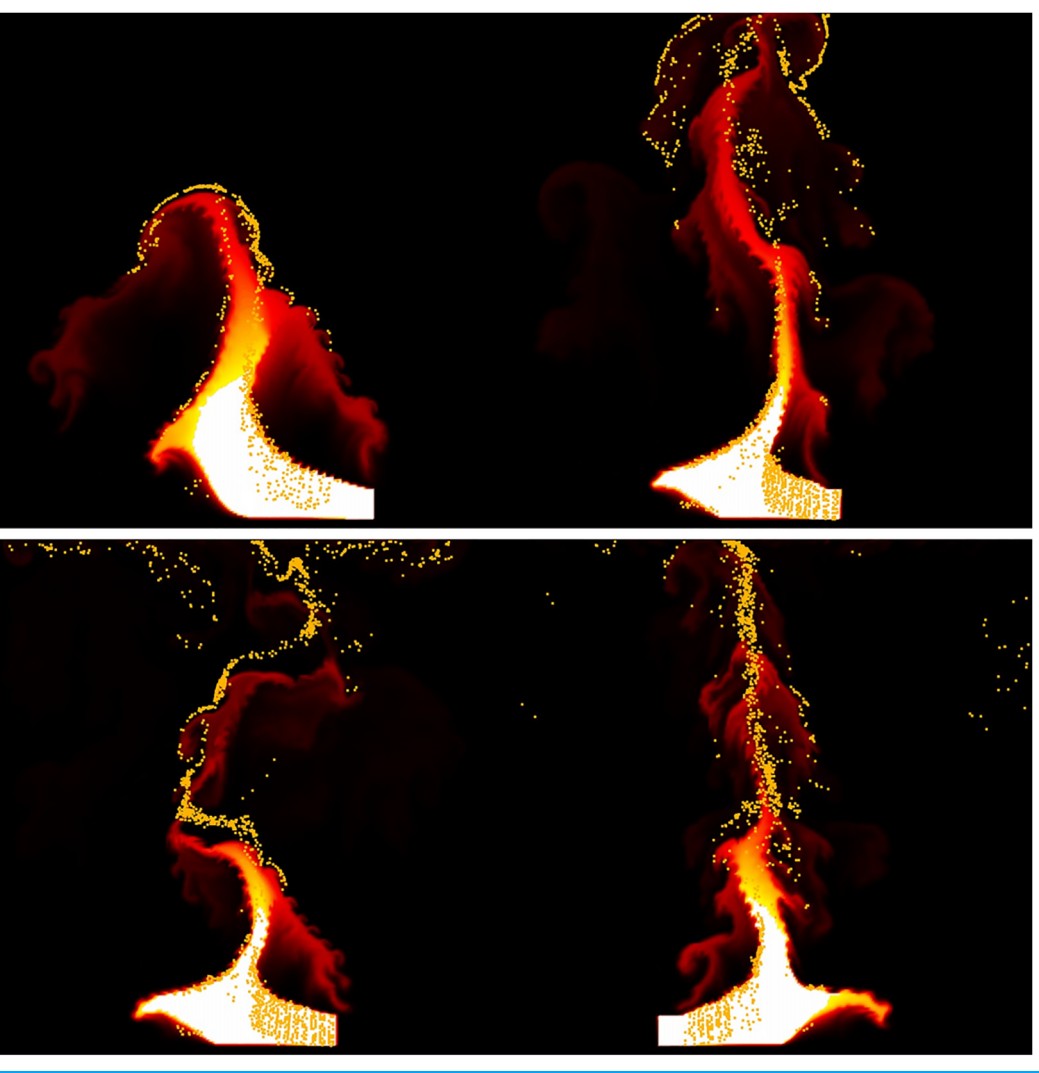

Figure 12 **Moving box-shaped flame.**               

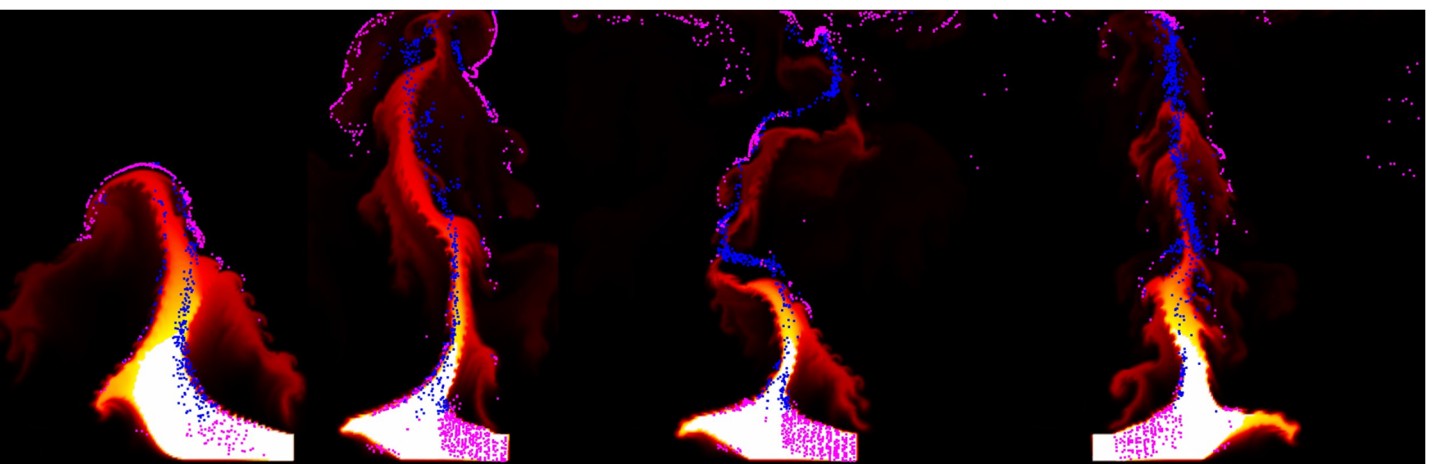

Figure 13 **Fire-flake particles with different movements in chaotic region.**               

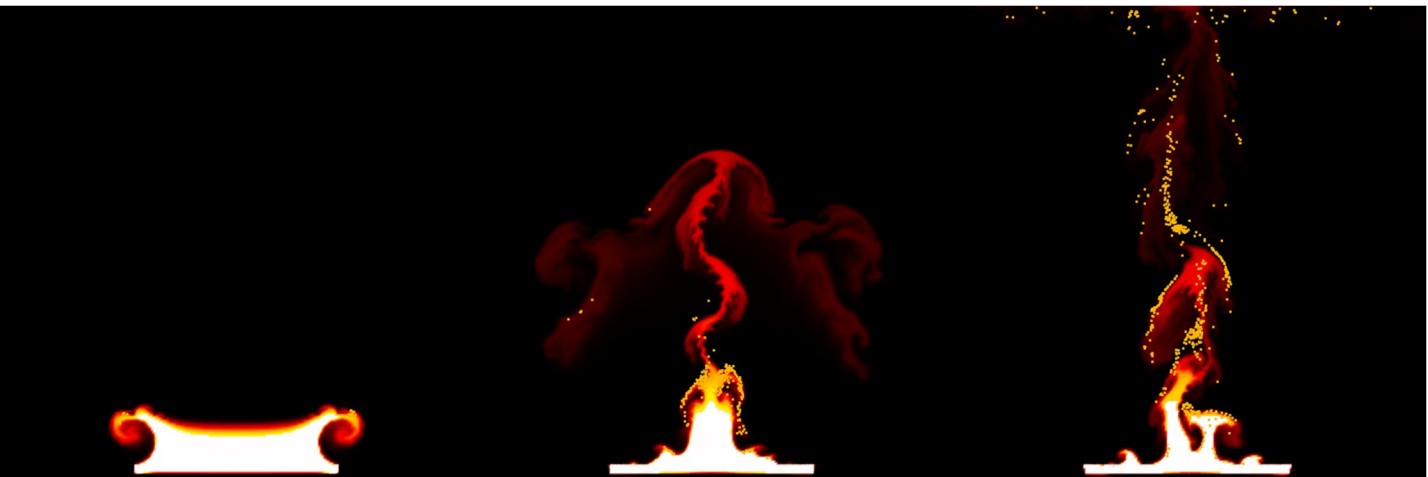

**Figure 14 Fire-flake particles from thin flame.**               

turbulent motion and consequently causing the fire-flake particles to move with less dispersion. The method proposed in this article does not unconditionally disperse fire-flakes. Instead, it considers the shape of the flame. This characteristic enables the generation of realistic fire-flake particle movements in various scenes.

Figure 15 depicts a rising flame interacting with obstacles. In this scene, we aim to observe two key features: 1) The movement of fire-flake particles when the temperature drops rapidly, and 2) the movement of fire-flake particles due to the interaction between the flame and obstacles. After colliding with the rectangular obstacle, as the temperature of the flame decreases, the movement of most fire-flake particles is significantly influenced by the stochastic advection method **g**, which takes into account random walk behavior. In this scene, most of the fire-flake particles are dispersed into the air without being strongly influenced by the movement of the flame (see Fig. 15A). In the previous method, even though it's the same scene, fire-flake particles were hardly represented (see Fig. 15B). In the previous method, fire-flake particles were generated at the collision points between the flame and obstacles due to the impact, but they were subsequently confined to certain areas due to the continuous inflow of the flame. This characteristic was evident in both the movement and velocity of fire-flake particles (see Fig. 15B).

Figure 16 illustrates a scenario where flames are ignited from spherical-shaped fuel injected at random positions. In this scenario, we aim to observe the movement of fire-flake particles dispersing into the air as a result of the rapid injection of fuel over a short period. When fuel is injected from various positions, turbulent flows are generated not only in the area where the flame is located but also in its surroundings. As a result of these turbulent interactions, distinct chaotic regions become apparent. Our method demonstrated that even after the flame disappears, fire-flake particles maintain a dispersed pattern in the air (see Fig. 16A). In contrast, the previous method exhibited a behavior where fire-flake particles seemed to closely follow the flame, sticking to it (see Fig. 16B). In

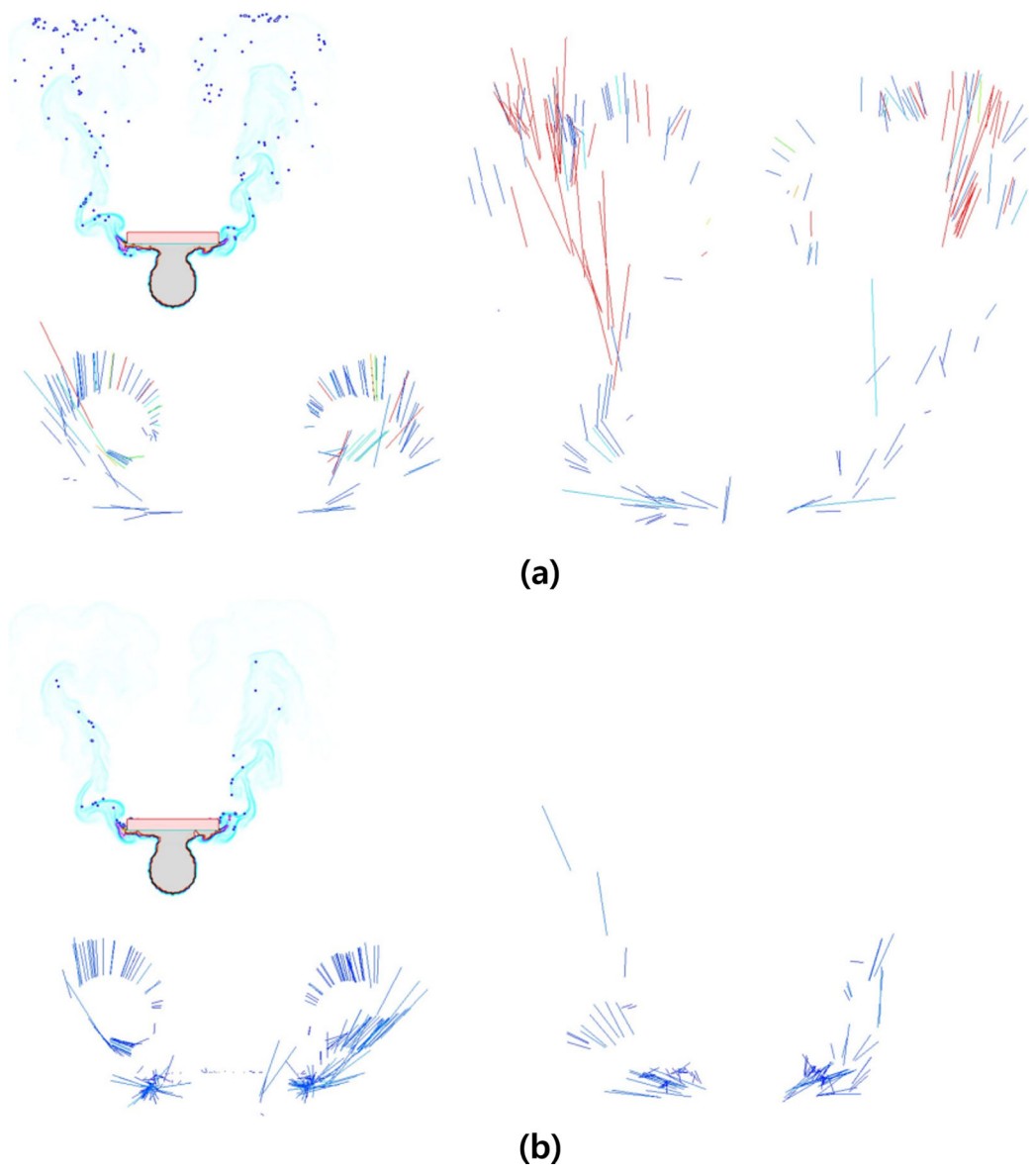

**(a)**

**(b)**

**Figure 15 Flames rising from interaction with obstacles (blue particle in inset image: fire-flake, line: velocity of fire-flake particle, frame number: 32, 70).** This figure has been subjected to contour filtering for clearer visualization of the results: (A) Our method, (B) Previous method.

the previous method, the movement of fire-flake particles relied solely on the movement of the flame, resulting in a lack of representation of intricate details within the chaos regions. Instead, the particles exhibited simplistic movements resembling a particle system. This behavior was also evident in the velocity of fire-flake particles (see Fig. 17B). However, our method captured the intricate movements of fire-flake particles using various buoyancy flows and stochastic advection techniques (see Fig. 17A). This capability enabled the representation of details that would be challenging to express solely through a simple random walk approach.

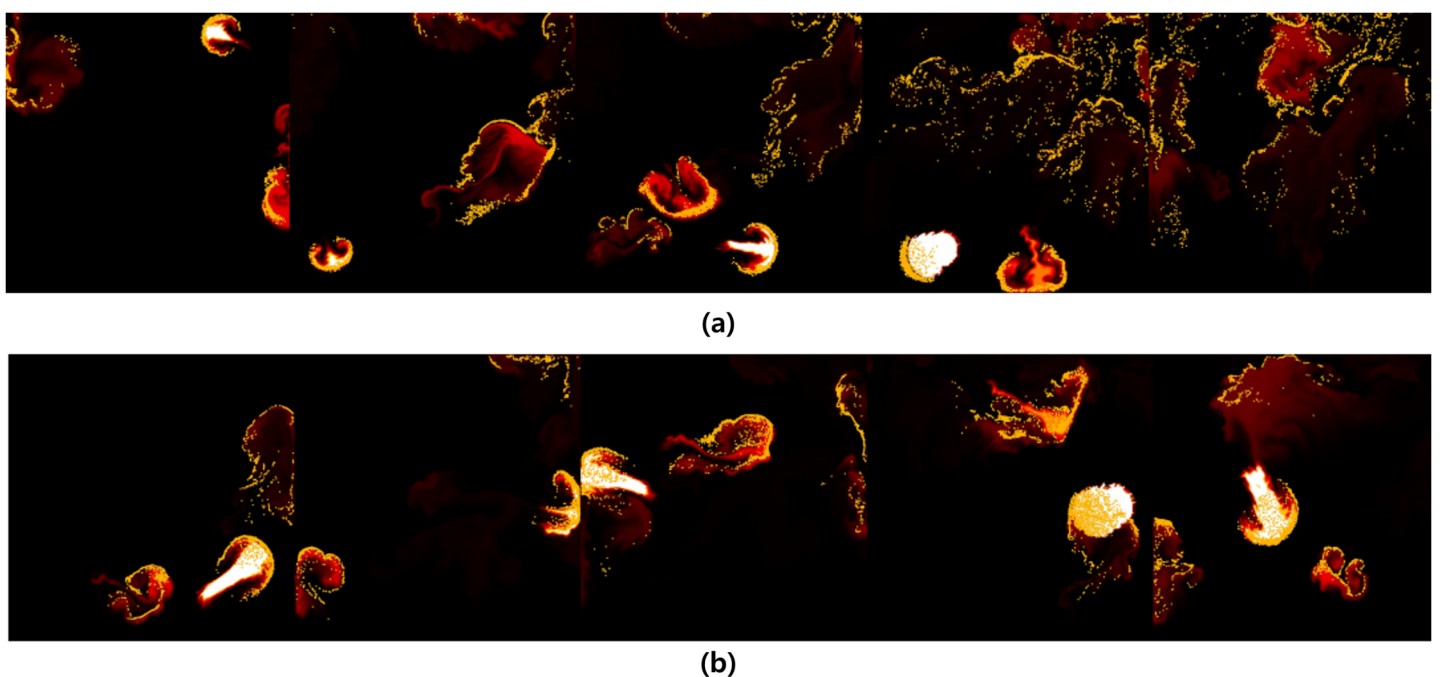

**Figure 16 Spherically shaped flame injected from random positions: (A) Our method, (B) Previous method.**

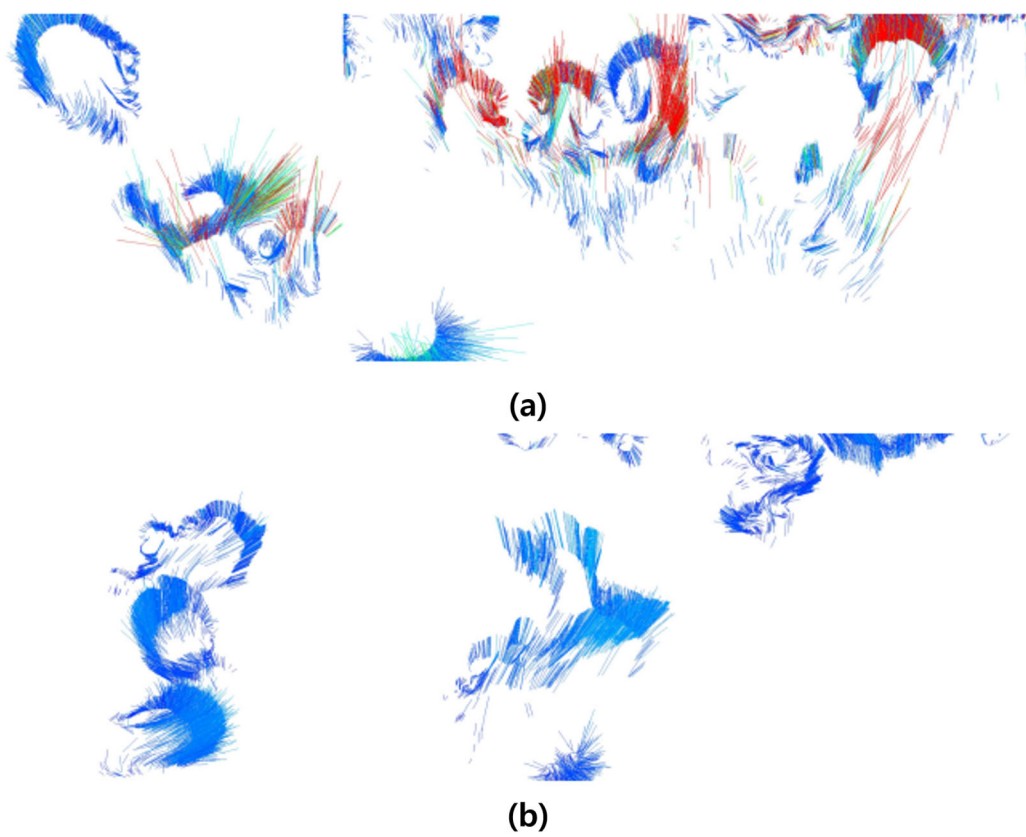

**Figure 17 Visualizing the velocity of fire-flake particles: (A) Our method, (B) Previous method.**

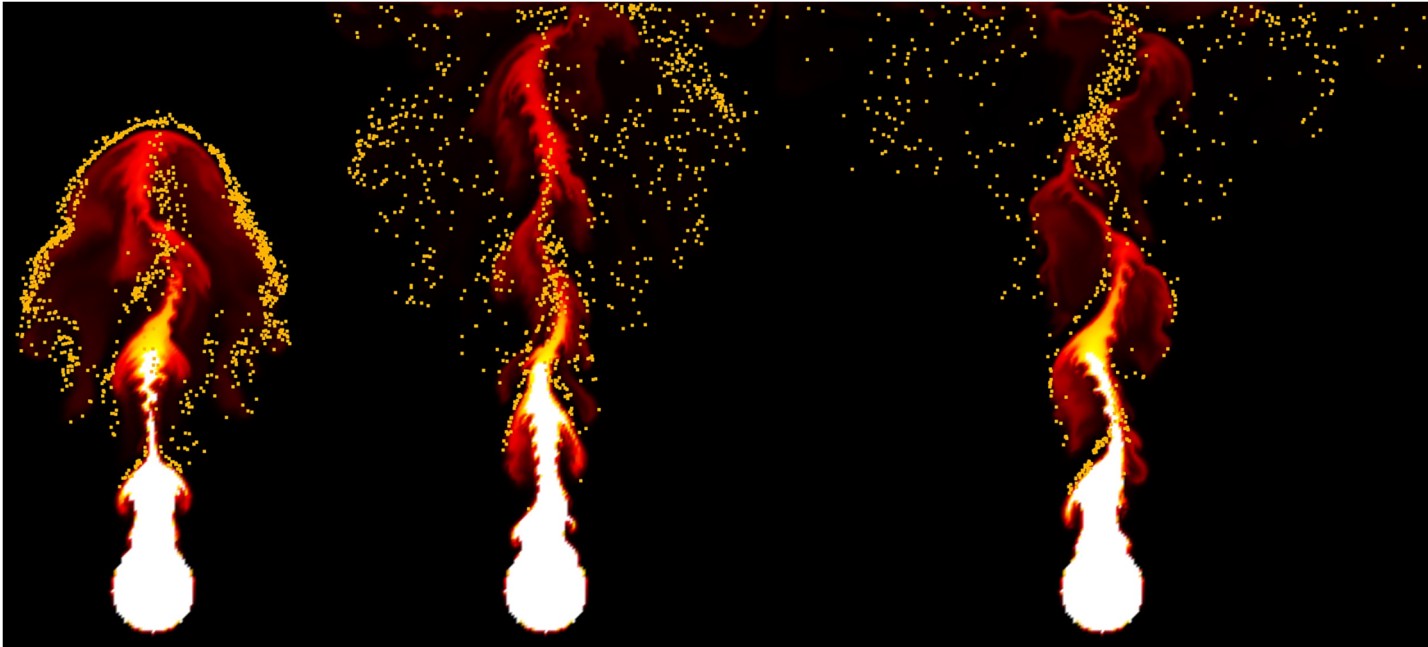

**Figure 18 The movement of fire-flake particles generated from the rising flame, learned through neural networks.**

## Learning results

In this section, we present the results generated by extending the simulation technique's solver with neural networks. We examine whether the results obtained through learning exhibit similar quality of representing fire-flake particles as the previously presented results.

Figure 18 shows the movement of fire-flake particles generated from the rising flame, learned through neural networks. Compared to the previous method where fire-flake particles tended to move closely to the flame (see Fig. 9B), the results obtained through AI-based learning effectively represented dispersed movement. When comparing the results obtained through our proposed simulation technique for representing fire-flake particles (see Fig. 9A) with the AI-based learning results, we can observe that the AI-based results generally exhibited a somewhat stronger dispersed flow. However, overall, the AI-based results closely resembled those achieved through the simulation technique.

Figure 19 illustrates the fire-flake flow learned through neural networks for the scene depicted in Fig. 12. In this scene, the complex fire-flake flow that emerges while rising is well represented, and it effectively captures the intricate movement of fire-flake particles in chaotic regions, similar to what is seen in Fig. 13. As observed in the previous result, this one also exhibits a more pronounced dispersed flow compared to the simulation technique. The enhanced dispersed flow of fire-flake particles through learning is attributed to the fact that the influence of **g** was strengthened, rather than simply dispersing all particles. In the Fig. 19, it's evident that there are fire-flake particles that are

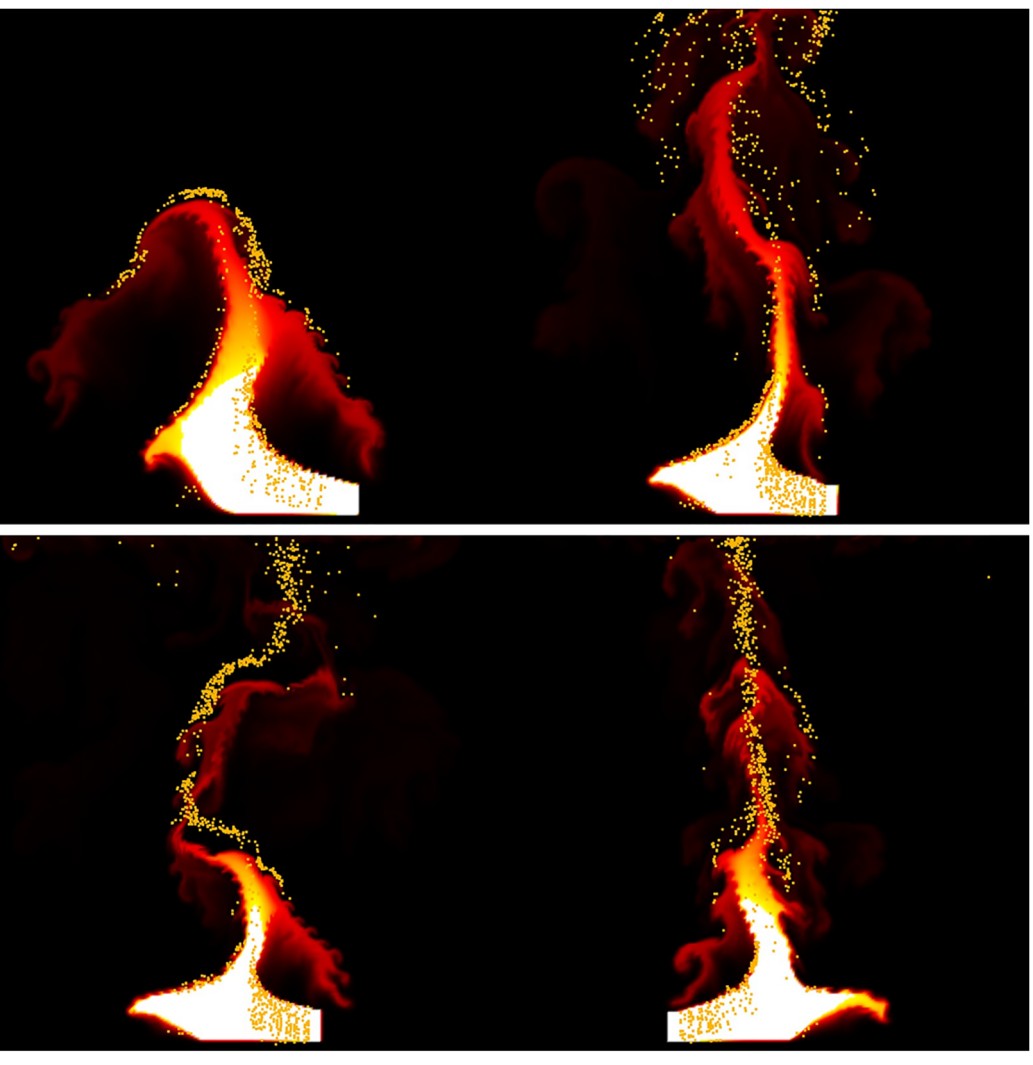

**Figure 19 The movement of fire-flake particles generated from moving box-shaped flame, learned through neural networks.**

strongly influenced by the flame, while other particles are dispersed without such direct influence. These results demonstrate that through learning, it's possible to represent not only fire-flake particles strongly influenced by the flame's movement but also particles dispersed in the air.

Figure 20 illustrates the fire-flake flow learned through neural networks for the scene depicted in Fig. 14. The generated fire-flake particles align well with the shape of the converging flame, accurately capturing both the fire-flake flow near the flame and the dispersion of particles into the air. The results of deleting fire-flake particles through learning with neural networks also appear to be handled naturally.

Figure 21 illustrates the fire-flake flow learned through neural networks for the scene depicted in Fig. 15. The representation of fire-flake particles dispersing according to the fire-flake flow, rather than clustering due to obstacles, is well captured in the results (see

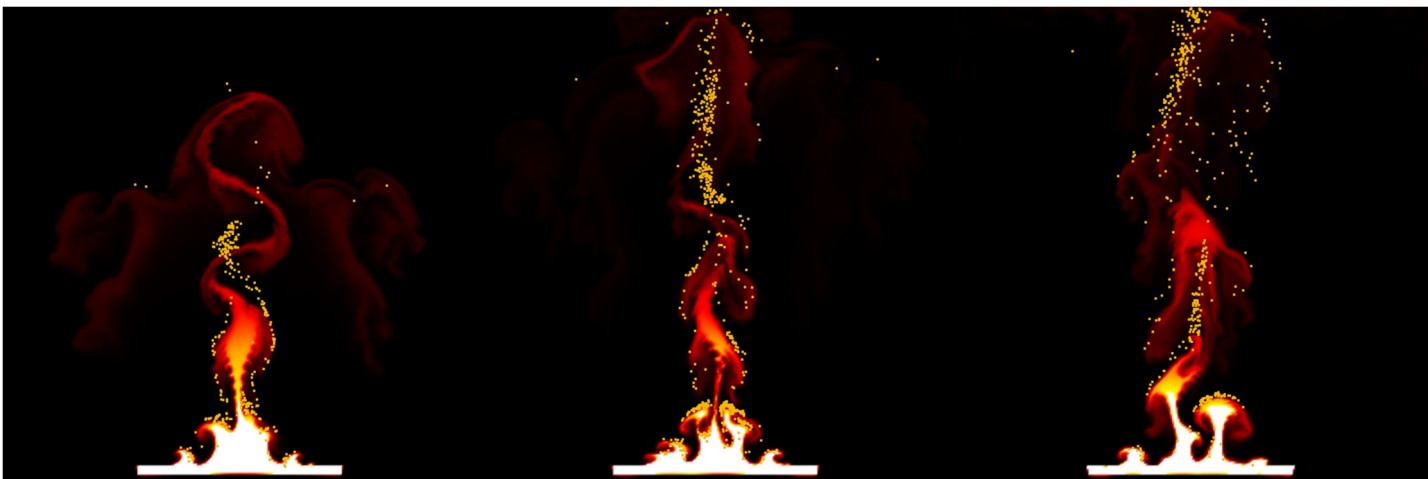

**Figure 20** The movement of fire-flake particles generated from blazing flame, learned through neural networks.

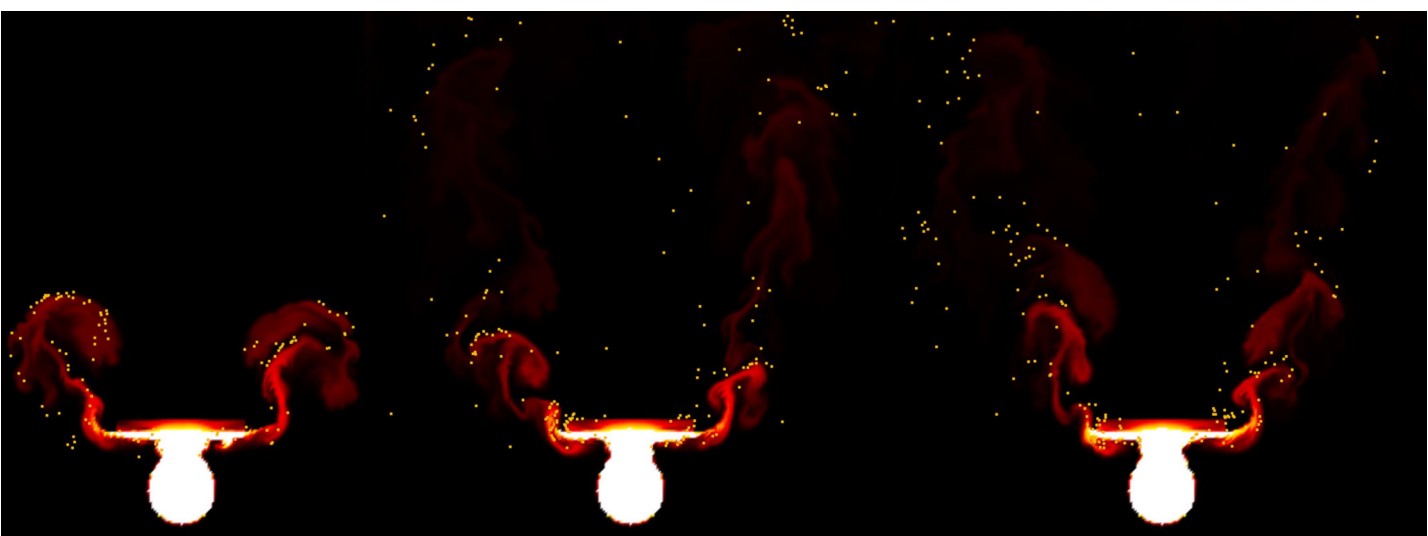

**Figure 21** The movement of fire-flake particles generated from blazing flame and solid interaction, learned through neural networks.

Fig. 15A). Unlike the previous method where fire-flake particles struggled to move properly due to clustering (see Fig. 15B), our method through learning presents natural movements similar to the simulation technique (see Fig. 21).

Figure 22 illustrates the fire-flake flow learned through neural networks for the scene depicted in Fig. 16. The dispersed flow in chaotic regions, as well as the fire-flake flow influenced by the shape of the flame, are both accurately and naturally represented in our results.

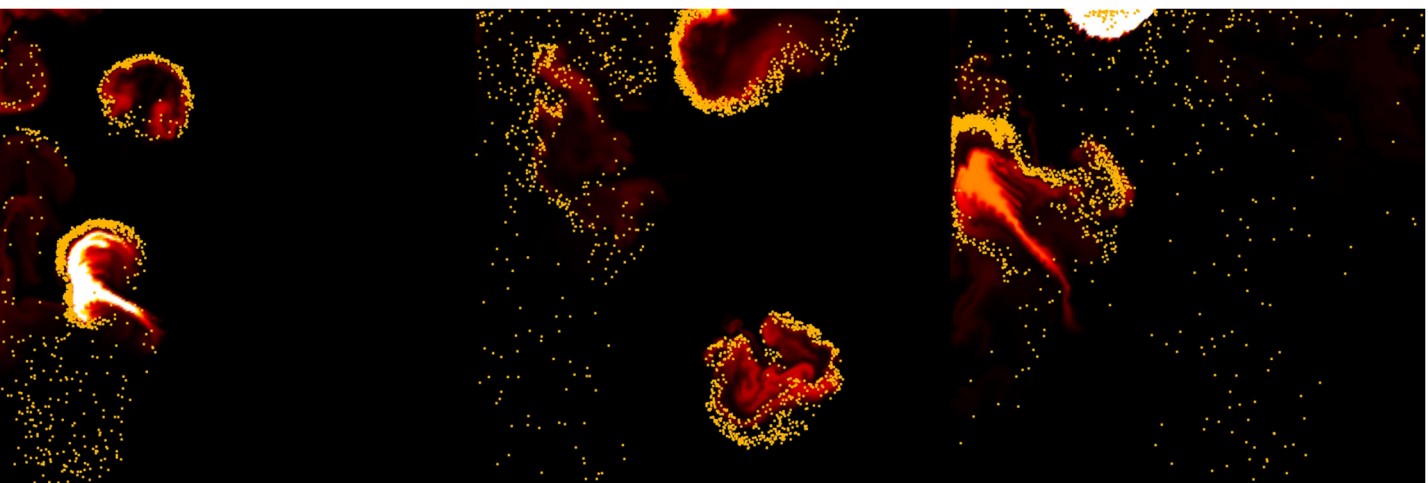

**Figure 22 The movement of fire-flake particles influenced by the interaction between solid objects and the flame, learned through neural networks.**

The generator network for fire-flake particle generation consists of three layers with 15, 5, and 2 nodes, respectively. In contrast, the advection network comprises four layers with 30, 24, 12, and 3 nodes. Both networks use the ReLU activation function and apply weight decay. The ADAM optimizer was used, and each network was trained independently. The learning rate was set to $10^{-5}$ for the generator network and $10^{-4}$ for the advection network. Each network was trained with 50,000 batches, using 10,000 training steps for the generator network and 300,000 training steps for the advection network.

The hyperparameters used in this study were not optimized through network training but were empirically tuned. Additionally, the reason for proposing fire-flake particle motion not only through numerical simulation but also *via* learning representation was to demonstrate that our method can be easily used without requiring expertise in complex numerical analysis.

## DISCUSSION AND SCALABILITY

In this section, we perform a comprehensive comparative analysis of our proposed method from various perspectives and assess its scalability and efficiency.

### Comparative experiments

#### Subgrid dynamics for representing fire-flake flow

In terms of enhancing subgrid details through a stochastic solver, our method is similar to the bubble dynamics proposed by *Kim, Song & Ko (2010)*. In their approach, to express the zigzag patterns of bubbles, they introduced a random walk-based motion influenced by buoyancy for adding details. However, the trembling motion aimed at lively representation of bubbles differs somewhat from the characteristics of fire-flake flow.

Figure 23 presents a comparison of the movement of dispersed bubble flow based on the dynamics of previous research and the movement of fire-flake flow using our method. The

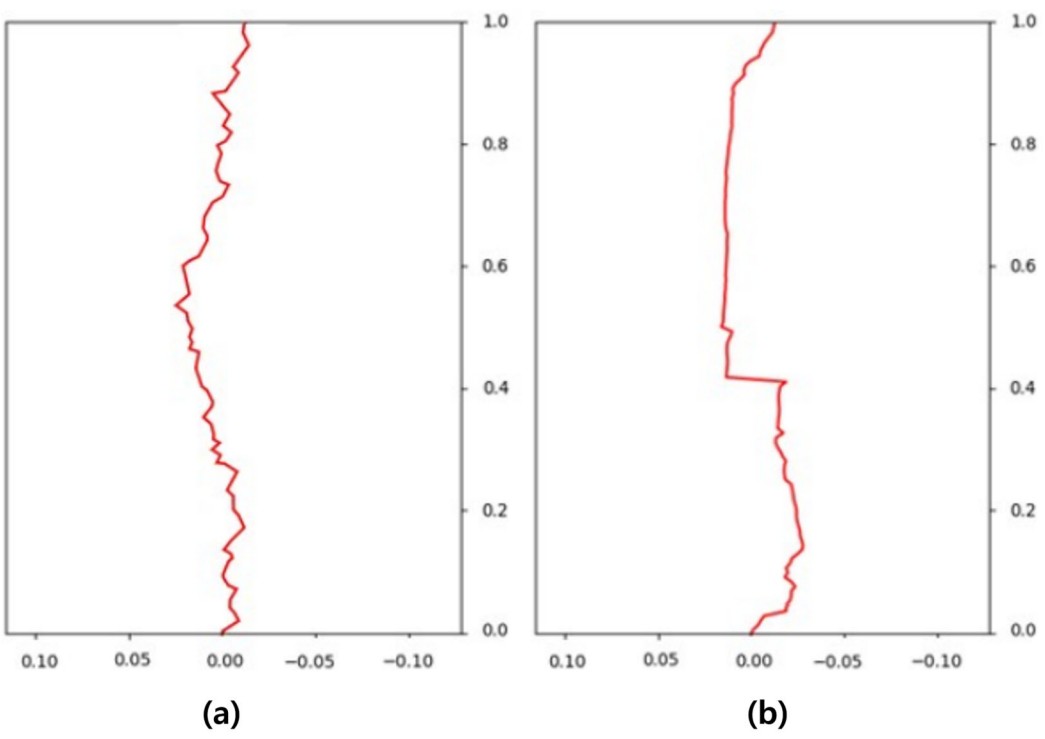

**Figure 23 Comparison of the dispersed flow generated by the previous method and ours: (A) Our method, (B) Previous method.**

motion of particles was tracked under the assumption that they rise upward. In the previous method, to represent the zigzag motion of a particle, a user-defined angle of $\theta$ was specified for its path to twist by, and this characteristic is also well evident in the particle path (see Fig. 23A). *Kim, Song & Ko (2010)* utilized a stochastic solver instead of considering interactions between bubbles, but to properly calculate this, it would have been necessary to model forces related to bubble dynamics, such as drag force, lift force, vorticity, attraction force, *etc.*, as originally proposed by *Hong et al. (2008)*. However, this method involves a significant computational burden and requires a considerable amount of time for parameter tuning to achieve stable simulations. *Kim, Song & Ko (2010)* efficiently captured the zigzag pattern of bubbles by employing a stochastic solver alone to address this issue (see Fig. 23A). However, applying this path to fire-flake flow results in a noisy motion due to the inherent jittery sensation in the path. On the other hand, our method effectively captures the dispersed fire-flake flow that naturally rises due to air resistance (see Fig. 23B).

### Movement of fire-flake particles through neural networks

In the previous method of *Choi et al. (2021)* that aimed to represent fire-flake particles using artificial intelligence, the results resembled a simple particle system with vertical ascension, considering only buoyancy. On the other hand, our method exhibits the characteristics of dispersed fire-flake particles based on the movement, shape, and momentum of the flame.

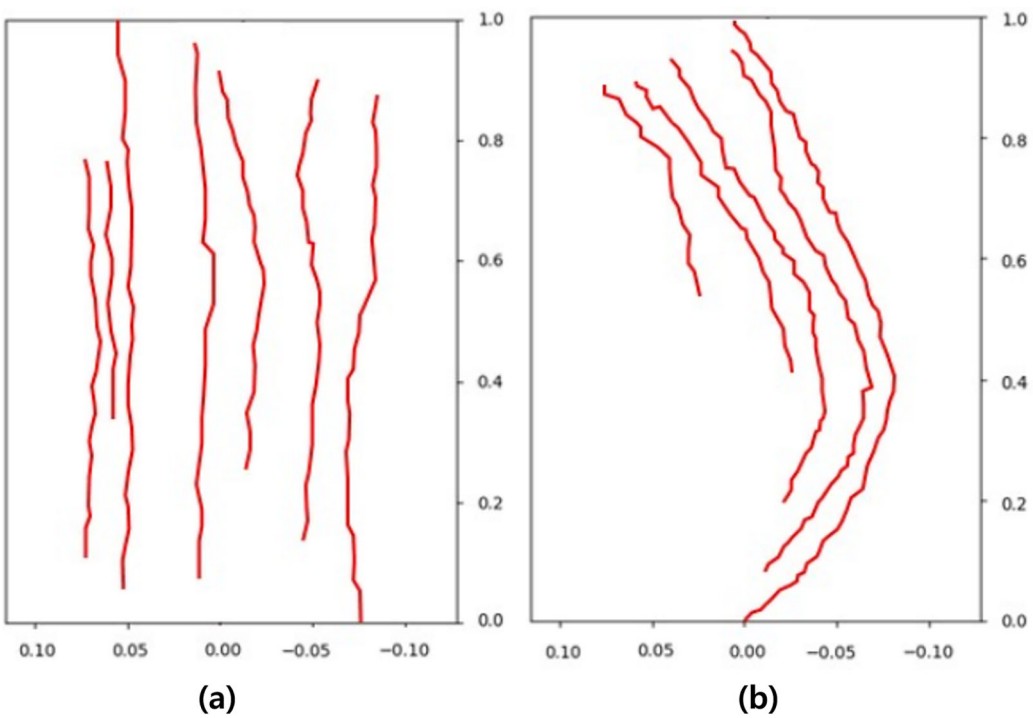

**Figure 24 Trajectories of fire-flake particles with previous method** *Choi et al. (2021)*: **(A) Blazing flame, (B) Moving box-shaped flame.**

Figure 24 shows the trajectories of several fire-flake particles generated using *Choi et al.*'s *(2021)* method. This method was unable to capture the complex movements occurring in chaotic regions of scenes where the flame rises or the emitter moves, as well as the dispersed fire-flake flow patterns. Unlike our method, which represents both the movements dependent on the flame and the dispersed movements in the air, their method mostly depicted simple upward movements with a small-scale zigzag pattern in some cases.

## Viscous clustering appearing from fire-flake particles

In contrast to the previous approach, the fire-flake particles represented in this study exhibit viscous clustering. These characteristics provide rich visual details when expressing various movements in chaotic regions. Generally, it is known that temperature is proportional to the vibrational activity of molecules constituting a substance. Similarly, hot air is associated with more vigorous motion, while cold air tends to have relatively less movement. In this study, these characteristics are considered by controlling buoyancy based on the temperature and lifespan of fire-flake particles. As a result, the motion of fire-flakes varies with changes in temperature, leading to distinct behaviors. The experimental results show that at high temperatures, a viscous clustering pattern emerges, while at relatively low temperatures, a dispersed flow pattern is observed. Although viscous clustering is not based on a strict physical theory, it has proven helpful in enriching the visual representation with intricate details.

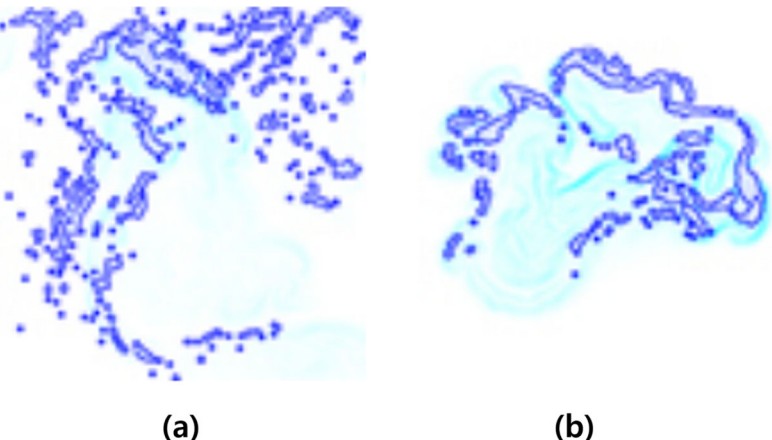

**(a)** **(b)**

**Figure 25** **Comparison of fire-flake particle movements based on temperature differences (blue: fire-flake particles, cyan: flame).** This figure has been subjected to contour filtering for clearer visualization of the results: (A) Dispersed flow (low temperature), (B) Flame-dependent flow (high temperature).

Figure 25 captures a portion of fire-flake particles represented by randomly injected flames. In the low-temperature region, the momentum of the flame weakened, leading to a dispersed flow that is more influenced by air resistance rather than flame motion. As seen in Fig. 25A, the pattern of dispersed fire-flake particles is well represented even in the filtered results. On the other hand, at high temperatures, the relatively vigorous momentum of the flame affects the fire-flake particles, causing them to be advected around the flame. The fire-flake particles depicted in this vicinity are significantly influenced by the flame, resulting in a viscous clustering formation rather than dispersion. They exhibit relatively fast movement due to the strong advection by the flame (see Fig. 25B).

## Integration with fire-flake texture method

One of the prominent methods to efficiently represent fire-flake particles is by analyzing the flow from videos to generate fire-flake particle textures. By using this approach, even without extensive knowledge of complex numerical analysis, it's relatively easy to express fire-flake effects effectively. The method proposed in this article can also be integrated into the image-based framework suggested by *Kim & Lee (2019)*. In their method, they computed a feature vector $\mathbf{Fv}^*$ using a DoG (Difference of Gaussian) filter-based approach to determine the direction in which the flame was burning and the direction of buoyancy-induced upward motion from the images. Furthermore, they used these calculated values as inputs to compute the fluid flow using the 2D Navier-Stokes equation, thereby representing the fluid flow. By integrating our method into this process, it becomes possible to easily represent individual fire-flake particles using neural networks, without the need for individual advection of each particle.

Figure 26 represents fire-flake particles that have been expressed by combining our method with the previous method. Our method not only introduces simulation techniques but also incorporates learning techniques to represent fire-flake particles. Furthermore, it

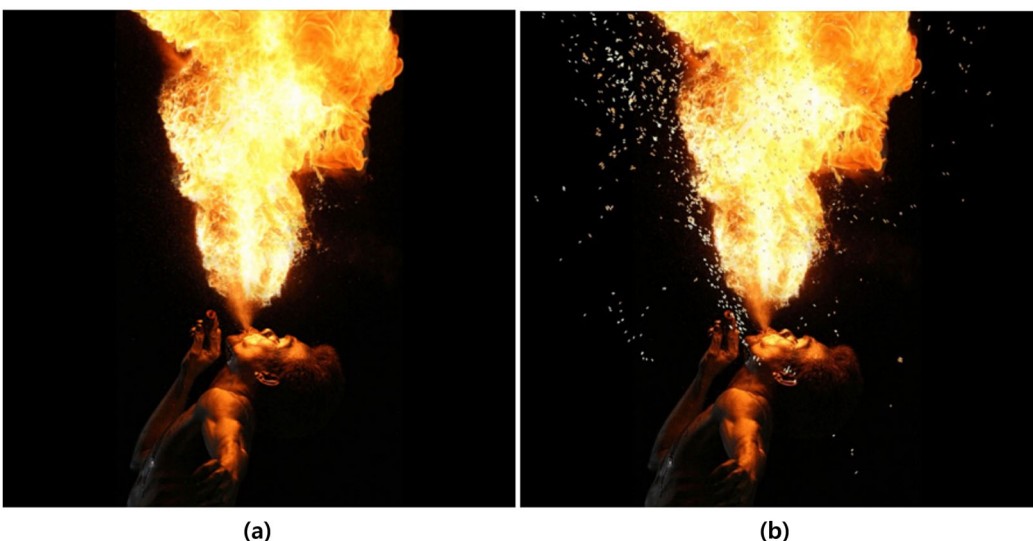

(a)                             (b)

**Figure 26 Synthesized fire-flake particles by integrating our method with Kim and Lee's method** *Kim & Lee (2019)*: **(A) Input data, (B) Synthesized fire-flake particles.**

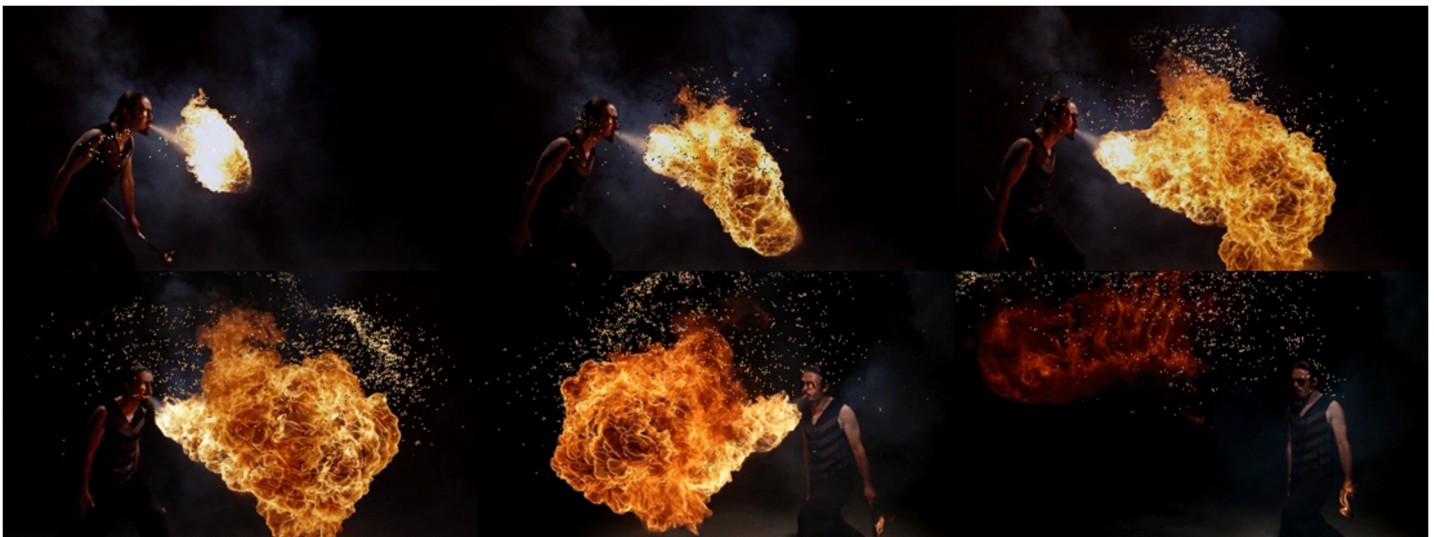

**Figure 27 Synthesized fire-flake particles by integrating our method with** *Kim & Lee*'s *(2019)* **method.**

can be easily integrated with previously proposed fire-flake texture methods, making it highly versatile and applicable. Figure 27 shows results obtained from experiments conducted in more dynamic scenario. The fire-flake particles are well represented in scenes where the actor emits flames, as well as in scenes where the camera is rotated, demonstrating a natural and smooth motion. The conditions for generating fire-flake

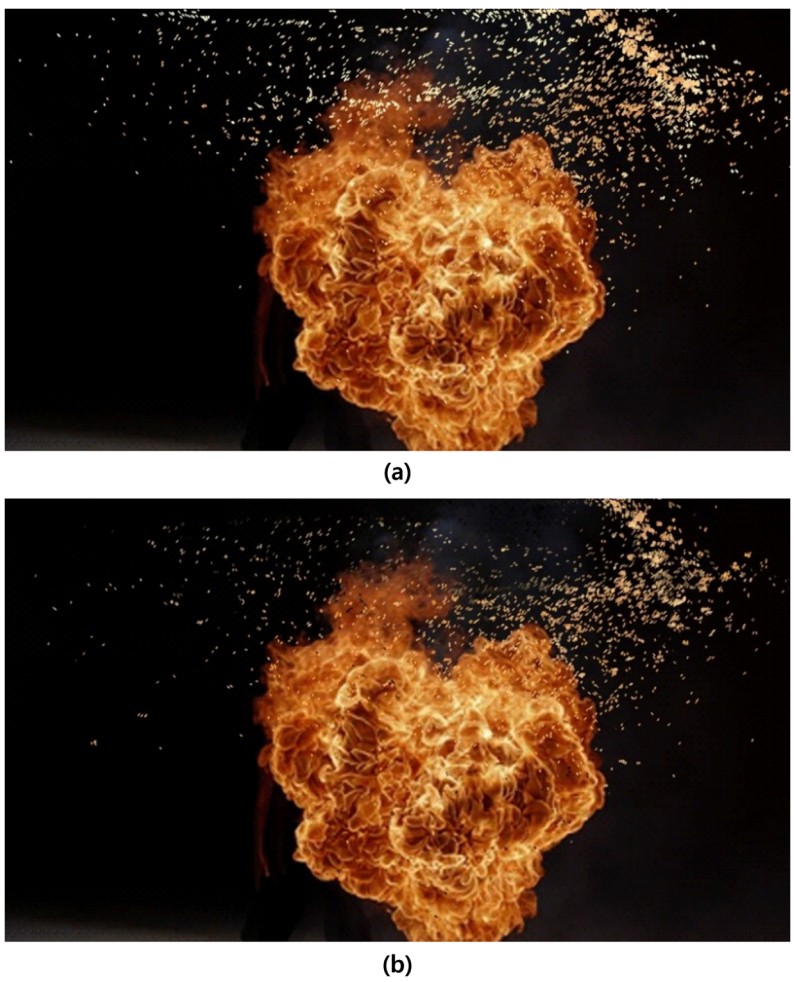

**Figure 28 Comparison of colors between fire-flake particles generated from the previous method and our method: (A) Previous method, (B) Our method.**

particles in Figs. 26 and 27 were based on the previous method, but their motion was represented using our method's neural networks for learning.

The generated fire-flake particles produce high-quality motion from the video input. However, in cases where there is a noticeable difference in color between the input data and the fire-flake particles, the synthesized result can sometimes appear unnatural (see Fig. 28A). In this article, to mitigate such discrepancies, the color of particles is calculated during rendering as follows:

$$(1 - \varsigma)I_\Gamma(\mathbf{v}_i) + \varsigma I_o(\mathbf{p}_i), \tag{22}$$

where $I_o$ represents the color at the location where $\mathbf{p}_i$ is situated in the input data, and $I_\Gamma$ is the interpolated color value based on $\mathbf{v}_i$ a predefined RGB color table: (0,0,0), (1.0,0.82,0.54), (0.88,0.62,0.4), RGB (0.98,0.68,0.3). Moreover, $\varsigma$ serves as both an interpolation weight and a representation of the lifespan of the fire-flake particle. Its range has been refined to lie between 0 and 1 for effective utilization. Consequently, the above

**Table 1 Experimental scenes (simulation and learning results).**

| Figure | Grid res. | Avg. number of fire-flake particles | Time-step |
|---|---|---|---|
| Fig. 9A | 256 × 256 | 960 | 0.1 |
| Fig. 9B | 256 × 256 | 738 | 0.1 |
| Fig. 12 | 256 × 256 | 1,421 | 0.1 |
| Fig. 14 | 256 × 256 | 733 | 0.1 |
| Fig. 15A | 256 × 256 | 148 | 0.1 |
| Fig. 15B | 256 × 256 | 198 | 0.1 |
| Fig. 16A | 256 × 256 | 4,103 | 0.1 |
| Fig. 16B | 256 × 256 | 2,494 | 0.1 |
| Fig. 18 | 256 × 256 | 960 | 0.1 |
| Fig. 19 | 256 × 256 | 1,421 | 0.1 |
| Fig. 20 | 256 × 256 | 733 | 0.1 |
| Fig. 21 | 256 × 256 | 148 | 0.1 |
| Fig. 22 | 256 × 256 | 4,103 | 0.1 |

equation efficiently renders fire-flake particles without temperature attributes by retrieving colors from the color table for particles with shorter lifespans and from the input data for particles with longer lifespans (see Fig. 28B).

## Reasons for representing fire-flake particles in 2D simulations

This article introduces a novel method to enhance the movement of fire-flake particles. Our method offers the capability to represent fire-flake particles not only through simulation but also through learning, allowing for potential applications in various fields. Especially, it can be utilized in areas such as games, VR/AR, and the metaverse to model realistic environments. As previously mentioned, we have integrated techniques that are more efficient for real-time content utilization rather than 3D simulations, and demonstrated the results of this integration. Since we have compared and analyzed the improved movement of fire-flake particles in 2D, we anticipate that these advantages will also be effectively represented in 3D scenarios.

## Accuracy of the generated positions of fire-flake particles

In reality, most fire-flakes are a type of byproduct that has been ejected due to the interaction between fire and solid objects. However, in this study, fire-flake particles were generated in conjunction with the momentum of the flame, which results in some differences from the actual phenomenon. While the design of fire-flake generation conditions wasn't based on direct physical phenomena, the intention was to enhance visual quality, making it applicable to various real-time performance-oriented content industries.

## Results analysis

Parameters and configurations used in this article are summarized in Tables 1 and 2, respectively. The configuration for the results obtained by integrating the fire-flake texture method with our method is summarized in the Table 3.

**Table 2 The result of integrating the previous method _Kim & Lee (2019)_ with our method.**

| Figure | Grid res. | Num. of input images | Avg. number of fire-flake particles | Avg. computation time (frame/sec) |
|--------|-----------|----------------------|-------------------------------------|-----------------------------------|
| Fig. 26 | 100 × 100 | 18 | 2,714 | 0.2 |
| Fig. 27 | 100 × 100 | 54 | 2,908 | 0.2 |

**Table 3 List of symbols used in the article.**

| Name | Description |
|------|-------------|
| $\mathbf{p}, \mathbf{v}, \mathbf{f}$ | Position/Velocity/Force of fire-flake particle |
| $m, T$ | Mass/Temperature of fire-flake particle |
| $\mathbf{u}, p, \rho$ | Velocity/Pressure/Density of the grid (flame) |
| $\mu, \mathbf{f}$ | Viscosity/Force of the grid (flame) |
| $\mathbf{f}^{air,drag,lift,buoy}$ | Airflow/Drag/Lift/Buoyancy force |
| $\mathbf{f}^{buoy\prime}$ | New buoyancy model |
| $k_{air}$ | Airflow coefficient |
| $k_{drag}$ | Drag coefficient |
| $k_{lift}$ | Lift coefficient |
| $k_{buoy}$ | Buoyancy coefficient |
| $\mathbf{f}_{buoyancy}$ | Upward normal vector |
| $E^t$ | Thermal energy |
| $k_t$ | Heat capacity factor |
| $E^k$ | Kinetic energy |
| $k_t$ | Thresholds for temperature |
| $k_e$ | Thresholds for energy change |
| $\mathbf{C}$ | Generating region of fire-flake particles |
| $\eta$ | Lifespan of fire-flake particle |
| $\varepsilon$ | Lifespan of fire-flake particle |
| $\mathbf{f}^{vorticity}$ | Particle-based vortex forces |
| $s$ | Probability function for fire-flake distribution |
| $\mathbf{g}$ | Probabilistic constant based on Gaussian random number |

As shown in Table 1, there was a slight difference in motion between simulation results and learning results, but there was not a significant difference in the number of generated fire-flake particles (see Fig. 30). In most of the results, it was observed that in the initial stages of the scenario, the number of fire-flake particles increased gradually in response to the generation of the flame. Figure 16 depicts the fire-flake particles represented in the scene with the interaction between the obstacle and the flame. Particularly in this scene, interesting patterns related to particle numbers have emerged. In our method, the trend of increasing fire-flake particles due to collisions was shown, whereas in the previous method, a repetitive pattern of fluctuation was observed (_Kim et al., 2017_). In our method, collisions

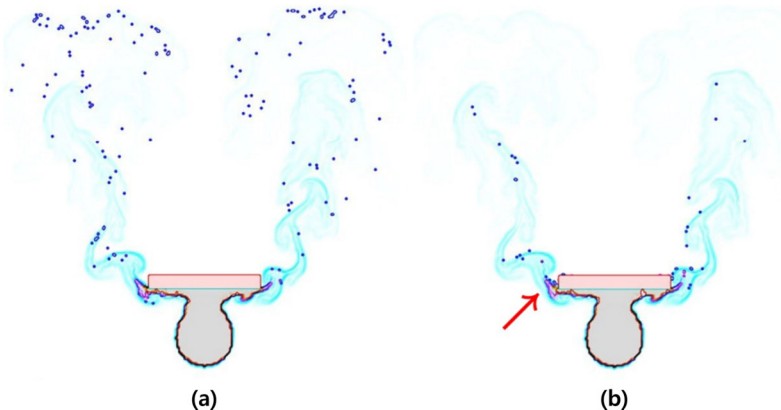

**Figure 29 Comparison areas where a fire-flake particle's movement becomes unstable (unstable area: red arrow): (A) Our method, (B) Previous method.**

led to a natural coupling and advection of fire-flake particles, whereas the previous technique resulted in particles being trapped around the obstacle (see Fig. 15B).

Figure 29 compared trapped fire-flake particles due to the interaction between the obstacle and the flame. Overall, it was observed that the fire-flake particles were generated and then quickly deleted as they became trapped, resulting in a decrease in their momentum. This phenomenon repeated cyclically. Examining the number of fire-flake particles, we can observe this pattern. For this reason, the chart shows a pattern where the number of particles doesn't simply increase but alternates between increases and decreases (see Fig. 16B chart in Fig. 30).

The fire-flake generation method proposed in this article can be applied not only to visual effects (VFX) expressing flames but also to various fields such as games and physics-based metaverse environment design. It is anticipated that this method can also be applied to small particles such as lightweight sand or dust that exhibit turbulent motion due to buoyancy or wind, as well as to air bubbles, foam, splashes, and other phenomena, which are lighter than air.

As observed from the presented results, our method consistently demonstrates visually improved outcomes when compared to previous methods. While our method does not rely on physical laws, it surpasses the limitations of previous methods by capturing the chaotic movement of fire-flake particles in a visually natural yet efficient manner. In contrast to previous methods where particle movement solely relied on underlying fluids, the proposed method effectively captures the intricate details of chaotic movement. This is achieved by accurately representing the dispersed fire-flake flow as it scatters through the air. Furthermore, in regions where the velocity field experiences abrupt changes due to obstacles, previous methods often resulted in fire-flake particles becoming trapped near objects. However, our method overcomes this issue and consistently represents these scenarios without particles getting trapped, ensuring stability and accuracy in the simulation. Moreover, our method offers visually improved simulation techniques while maintaining nearly identical computation times compared to the previous approaches.

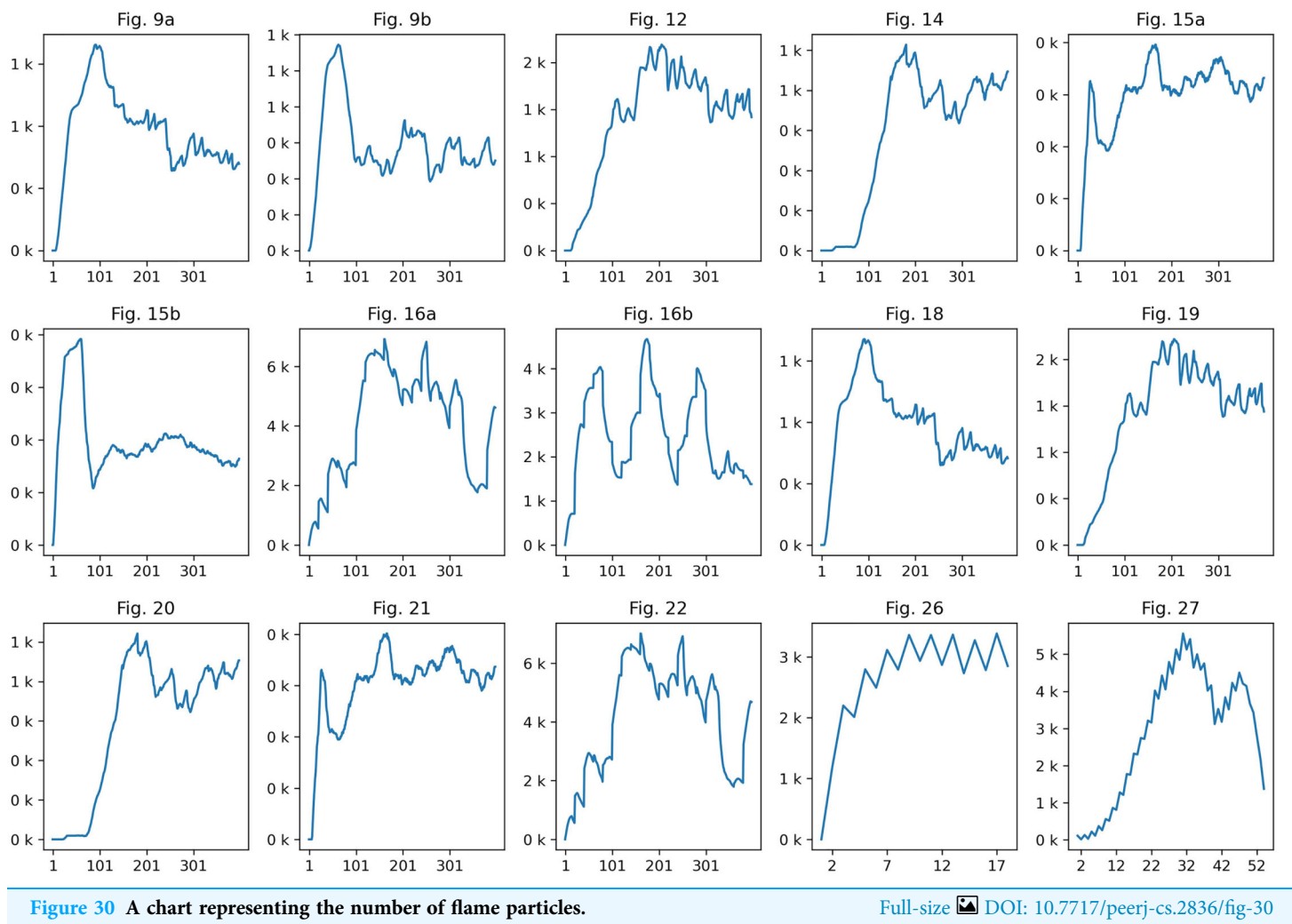

**Figure 30 A chart representing the number of flame particles.**

## Computational complexity

To the best of our knowledge, there is currently no precise method for calculating the chaotic motion of fire-flakes originating from flames. Even in the Navier-Stokes equations used to represent flames, fire-flake particles are depicted by generating turbulent flow based on advection, vorticity, or vortex particles corresponding to flame changes. The same approach applies to bubbles. As a result, the turbulent flow that should be represented by fire-flakes becomes dependent solely on the flame, making it insufficient for capturing complex chaotic motion. In particular, due to temperature variations caused by the flame, air resistance, and the extremely lightweight nature of fire-flakes, our method aims to efficiently represent the resulting chaotic motion and dispersed patterns under air resistance. This approach avoids traditional methods, such as using linear systems or iterative Gauss-Seidel calculations.

The proposed method's advantages include producing visually detailed fire-flake motion at a level close to visual simulation, and through extending the solver using learning representation, bypassing the complex computation processes to quickly generate

fire-flake particles. In this study, we used an Eulerian grid-based simulation to represent the flame, which results in a time complexity of $O(n^2)$, due to the need to iterate over each grid cell and calculate interactions with neighboring cells. The spatial complexity depends on grid resolution and is typically $O(n^2)$. With the addition of particle-based fire-flakes, the time complexity can range from $O(n^2)$ to $O(nlogn)$, depending on the number of particles and the grid resolution, considering the computations necessary for interactions among fire-flake particles and fluid flow simulation. The spatial complexity also depends on the grid resolution and the number of fire-flake particles, generally ranging from $O(n^2)$ to $O(n)$.

## Comparison with previous methods

In this section, we compare and analyze the proposed method with recent previous methods for simulating fire-flakes, which are secondary effects expressed by fire/flame simulation. The method by *Kim et al. (2017)* represents fire-flake particles within the flame using the dynamics of bubble simulation. They proposed advection influenced by flame and temperature by internally applying drag force, attraction force, lift force, and SPH-based vorticity confinement. However, it fails to capture the chaotic motion or dispersed patterns of fire-flakes, which is evident from comparisons in Figs. 9, 15–17, where our method demonstrates improved fire-flake motion in terms of visual simulation.

The method by *Choi et al. (2021)* extends *Kim et al.*'s *(2017)* approach using learning representation. They employed a simple structure that learns fire-flake particles through a neural network, but did not consider the number and proportion of fire-flake particles, which limited their ability to handle complex chaotic motion. Consequently, their results mainly show upward buoyant motion, lacking essential fire-flake characteristics. In contrast, our research shows complex and rich motions of fire-flake particles, especially in regions where chaotic motion and dispersed patterns should occur, making our method applicable across a wider range of fields.

The proposed method can be applied to VFX and visual simulation fields where detailed fire simulation is required. *Kim & Lee (2019)* introduced a framework that uses video or image sequence files as input data to infer the underlying flow and generate fire-flake effects. Combining our method with this approach would not only enhance the realism of fire-flake effects but also optimize the learning representation process. Additionally, since this approach relies on video or image input rather than 3D simulation, it eliminates the need for prior knowledge of complex numerical models, expanding its applicability. Furthermore, our method can be integrated with the still-frame simulation proposed by *Son et al. (2013)*. This technique simulates flame propagation along image boundaries but is limited to flame representation and does not account for secondary effects such as fire-flakes. By incorporating our approach, more dynamic and visually engaging image effects can be achieved.

Compared to existing fire-flake particle representation methods, our method showed minimal increase in computation time. This is because the number of fire-flake particles is relatively small compared to the underlying flame simulation, and no additional matrix operations are required. However, when using learning representation for fire-flake

particle modeling, the dispersed motion appeared more pronounced than in the simulation-based approach. This suggests the need for further refinement of the network design, highlighting an important research challenge for future work.

## Limitation

The proposed method approximates fire-flake motion based on a stochastic solver, which offers computational efficiency. However, it has several limitations. Since the fire-flakes are not generated from solid-flame interaction, their movement lacks physical accuracy. Consequently, the method cannot fully depict the detailed behavior of burning fragments being ejected from the solid, nor can it capture the vortices formed during the volume-loss process. Although adding a condition that assumes that fire flakes will be generated from the burning solid might allow for a similar representation, it complicates the procedure and still does not provide a physical approach. Furthermore, although the motion is chaotic, it depends on the flame and is insufficient to consider the multiphase flow induced by air.

In this study, we employed *Stam*'s *(1999)* method to approximate the Navier-Stokes equations for representing the underlying fluid flow. Internally, the Conjugate Gradient method was applied for pressure computation, while all momentum calculations, excluding fire-flake particles, were handled using the Staggered Marker-and-Cell method (*Harlow & Welch, 1965*). Additionally, flame-solid interaction was modeled using a variational framework (*Batty, Bertails & Bridson, 2007*), which enables fast and stable coupling. Since flames are generally represented as gases, we designed the underlying flow based on Eulerian fluid simulation using density and temperature. In particular, leveraging the fire effects proposed by *Son et al. (2013)* allows for efficient modeling of flame effects influenced by airflow while reducing computational costs. This approach is especially useful when detailed flame motion is required.

The proposed method is not a fully 3D simulation approach. While it is technically possible to extend the 2D Eulerian approach to 3D, the computational cost would increase exponentially, leading us to adopt an alternative approach in this study. *Kim & Lee (2019)* proposed a framework that infers the underlying flow from video or image sequence files and generates fire-flake effects based on this information. By integrating our method with this approach, we can achieve more realistic fire-flake effects while also reducing the computational load of the learning representation process. Additionally, since 3D simulation results can be converted into video data for input, we did not design our algorithm as a fully 3D simulation method. However, if a 3D simulation is essential, we recommend optimizing the learning representation process. Otherwise, a 3D CNN structure would be required, which is computationally inefficient. Instead, employing an adaptive octree-based CNN approach would be a more suitable alternative (*Wang et al., 2018*). Furthermore, this study does not consider interactions resulting from collisions between fire-flake particles. When fire-flake particles collide, they may exhibit sudden bouncing motions or unstable behaviors. To avoid such instabilities, we deliberately excluded collision-based interactions from our model.

## CONCLUSIONS AND FUTURE WORK

In this article, we have proposed an efficient technique to simulate dispersed fire-flake particles that respond to the movement of a flame, capturing chaotic fire-flake flow patterns in regions of chaos. Furthermore, we extended this approach to incorporate learning through neural networks. Unlike the previous approach that solely relied on the movement of the flame, our method improves the visual quality by introducing various forms of buoyancy and a novel advection method. This enhancement provides finer details and realism in the simulation. Unlike the typical random walk approach that adds noise randomly to the movement, our method takes into consideration the size and direction of the flame. This allows us to express fire-flake particles stably in most scenes without the need for parameter adjustments.

However, despite these advantages, there are several limitations in our study. We did not consider the interaction between fire-flake particles or the turbulence resulting from it, making it difficult to represent micro-level details or a large number of fire-flake particles. For example, aspects like collision, merging, shape deformation, and overlapping of fire-flake particles were not taken into account in this study. To represent such phenomena, implementing proximity tests or collision detection among a large number of fire-flake particles would be necessary. In this article, since direct interactions between fire-flake particles were not considered, there might be some unnatural aspects. However, since we are not dealing with phenomena where objects undergo significant shape deformations, such as the merging of air bubbles, this issue is not considered a critical concern. Indeed, calculating interactions between individual fire-flake particles could lead to a performance degradation in the algorithm, making it challenging to represent real-time content. Hence, in this article, we approximated the interactions by applying buoyancy and advection based on particle attributes, allowing for more efficient computation without sacrificing real-time rendering capabilities.

Typically, fire-flake particles are generated when solid fuel (carbon) is expelled from the body due to internal or external impacts on the fuel, causing it to become buoyant and float away. Therefore, strictly speaking, the location where fire-flake is generated should be at the boundary of the fuel, rather than at the interface between the flame and the air. The proposed method in this article aims to realistically represent the movement of fire-flake particles, and as mentioned earlier, the generation conditions are not entirely based on physics. Therefore, applying our method to techniques that represent fire-flakes detaching from solid surfaces could potentially enhance the realism of the fire-flake effect. Moreover, our method offers a framework that can be easily integrated with other techniques, further enhancing its versatility and applicability. In the future, we plan to not only address the aforementioned challenges but also explore methods to efficiently compute fire-flake particles using the temperature of the flame and its geometric characteristics, without the need for explicit simulation.

### Funding

This research was supported by Basic Science Research Program through the National Research Foundation of Korea (NRF) funded by the Ministry of Education (2022R1F1A1063180) (Contribution Rate: 50%). This work was supported by Institute of Information & communications Technology Planning & Evaluation (IITP) grant funded by the Korea government (MSIT) (No. RS-2022-00155915, Artificial Intelligence Convergence Innovation Human Resources Development (Inha University)) (Contribution Rate: 50%). The funders had no role in study design, data collection and analysis, decision to publish, or preparation of the manuscript.

### Grant Disclosures

The following grant information was disclosed by the authors:
National Research Foundation of Korea (NRF).
Ministry of Education: 2022R1F1A1063180.
Institute of Information & Communications Technology Planning & Evaluation (IITP).
Korea Government (MSIT): RS-2022-00155915.
Artificial Intelligence Convergence Innovation Human Resources Development (Inha University).

### Competing Interests

The authors declare that they have no competing interests.

### Author Contributions

- Jong-Hyun Kim conceived and designed the experiments, performed the experiments, prepared figures and/or tables, and approved the final draft.
- Jung Lee analyzed the data, performed the computation work, authored or reviewed drafts of the article, and approved the final draft.

### Data Availability

   The data are found in the Supplemental Files.

### Supplemental Information

Supplemental information for this article can be found online at http://dx.doi.org/10.7717/peerj-cs.2836#supplemental-information.

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
