# Peer review of "Numerical dispersed flow simulation of fire-flake particle dynamics and its learning representation"

_PeerJ Computer Science, doi:10.7717/peerj-cs.2836_

## Round 0.1 · original submission · Major Revisions

Please review and address all comments provided by the reviewers, with particular attention to the feedback from Reviewer 4.

Reviewer 1 ·

Basic reporting

This paper proposes an efficient technique to simulate dispersed fire-flake particles that respond to the movement of a flame. The paper is well-organized, and the findings are clearly represented. I would recommend the paper for publication after minor revision.
1. Check the comma and the punctuation in the text and at the ends of equations (Equations 3, 6 and the rest).
2. Check the equation number on line 215.
3. Tell us more about the numerical method used. What are the limitations of the method? Are there other robust methods? What motivates the choice of the current method to present the results.
4. The abstract should contain some obtained results through an ending sentence.
5. On Line 222, replace ‘In this paper’ by ‘In addition’ or any equivalent expression as it is written in the previous sentence.
6. Discuss the implications of your findings for real-world applications.

Experimental design

Yes

Validity of the findings

Yes

Additional comments

NO

Annotated reviews are not available for download in order to protect the identity of reviewers who chose to remain anonymous.

·

Basic reporting

The overall workflow of the work is appreciable.

Experimental design

Experimental design and the results provided is goo and sufficient.

Validity of the findings

The article meets the standard of the journal due to the experimental results given is good.

Reviewer 3 ·

Basic reporting

- The paper is well-structured and follows the standard conventions of PeerJ Computer Science.
- The writing is generally clear and professional. However, there are a few minor grammatical issues that could be improved for better readability. A proofreading pass would help refine the clarity of certain sentences.
- The literature review is thorough and provides a solid context for the research. However, the introduction could benefit from a slightly more explicit statement of how this study differs from prior works on fire-flake particle simulation.
- Consider briefly summarizing the key novelty of the proposed method in the introduction to differentiate it more clearly from previous approaches.

Experimental design

- The study presents a well-defined research question and describes the methodology in sufficient detail.
- The proposed numerical and learning-based methods are well-explained, making replication feasible.
- One aspect that could be clarified is the choice of specific hyperparameters in the AI-based learning component. While some values are given (e.g., thresholds, coefficients), a brief justification of their selection would enhance the methodological transparency.
- Consider adding a short explanation of how the hyperparameters were determined (e.g., empirical tuning, prior experiments, or theoretical justifications).

Validity of the findings

- The experimental results are well-presented with clear visualizations that support the conclusions.
- The comparison with previous methods is effective in demonstrating the advantages of the proposed approach.
- One minor suggestion is to clarify whether the proposed method introduces any trade-offs in computational cost compared to alternative approaches. While efficiency is mentioned, a brief note on potential computational overheads (if any) would be useful.
- Consider adding a short discussion of any computational trade-offs (e.g., runtime comparisons or memory usage) between the proposed method and prior techniques.
- While the paper highlights the advantages of the proposed method, a short discussion on its limitations (e.g., specific cases where performance might degrade) could help provide a balanced perspective.

Additional comments

- Overall, this is a strong paper with a well-structured presentation and meaningful contributions.
- The minor revisions suggested mainly focus on improving clarity, readability, and methodological transparency rather than fundamental issues.
- With minor refinements, this paper will be in great shape for publication.

·

Basic reporting

The paper is well-written and easy to follow. However, it would benefit from more emphasis on the rationale behind the development of the proposed method. While the paper thoroughly explains "what" was done, the reasoning behind the specific design choices remains unclear. Further clarification on this would strengthen the work. See below for additional details.

Experimental design

To summarize the contribution, the paper adapts the method from Kim et al. (2010), originally designed for simulating small bubbles in water, and applies it to simulate fire flakes using a relatively simple fire model from Nguyen et al. (2002). This contribution seems somewhat minimal, especially considering the length of the paper.

The authors make several claims about why the motion of fire flakes is fundamentally different from that of bubbles:
50: "On the other hand, fire-flake particles, being lighter than air and significantly influenced by air resistance, exhibit relatively more chaotic and disorderly movement."
135: "Most existing methods have focused on representing bubbles in water, and there has been relatively less research specifically dedicated to modeling fire-flake particles represented by flames."
253: "unlike air bubbles in water, near the flame, there are turbulent interactions between fuel and air that give rise to chaotic movements."
447: "Unlike air bubbles, fire-flake particles exhibit highly dynamic movements in chaotic regions."
However, these assertions are not supported by evidence (citations would be helpful), and yet, they still proceed to apply an existing bubble simulation technique to model fire flakes, which appears contradictory.

Additionally, the paper proposes training a neural network to learn the emission, trajectories, and lifespan of fire flakes. However, the training uses synthetic data generated by the paper's own simulation method, which raises the question of why an already established model is being "re-learned."

Validity of the findings

The main result of the paper shows that the proposed method generates more break-up in the motion of fire flake particles compared to previous methods that did not incorporate chaotic advection. While this outcome is expected, it is also unsurprising and aligns with the findings of Kim et al. (2010), who achieved something similar with bubbles.

The machine learning (ML) aspect of the paper is intriguing, but as noted earlier, its purpose remains unclear. The approach essentially re-learns an already established model (which, not surprisingly, yields similar results), and the paper does not demonstrate any performance benefits of using an ML technique over traditional simulation methods.

Additional comments

All of the simulations in the paper are conducted in 2D, and there is no supplemental video provided, making it difficult to evaluate the quality of the results.

Overall, I would consider the paper's contribution to be quite limited. It seems premature for publication and would require significant revisions.

---

## Round 0.2 · accepted · Accept

After reviewing the revised manuscript and carefully considering the reviewers' comments, I confirm that the authors have addressed all the concerns raised. As the editor, I have thoroughly assessed the revisions and am satisfied with the current version. I believe this manuscript is now ready for publication.